**Data Availability Statement:** All data are publicly available. Specific data related to this paper appear in the Supporting Information files.

**Funding:** This project was funded in whole or in part with Federal funds from the National Cancer Institute, National Institutes of Health, under

# Bioinformatic analysis linking genomic defects to chemosensitivity and mechanism of action

**David G. Covell** ***

Information Technologies Branch, Developmental Therapeutics Program, National Cancer Institute, Frederick, MD, United States of America

* covelld@mail.nih.gov

## Abstract

A joint analysis of the NCI60 small molecule screening data, their genetically defective genes, and mechanisms of action (MOA) of FDA approved cancer drugs screened in the NCI60 is proposed for identifying links between chemosensitivity, genomic defects and MOA. Self-Organizing-Maps (SOMs) are used to organize the chemosensitivity data. Student's t-tests are used to identify SOM clusters with enhanced chemosensitivity for tumor cell lines with versus without genetically defective genes. Fisher's exact and chi-square tests are used to reveal instances where defective gene to chemosensitivity associations have enriched MOAs. The results of this analysis find a relatively small set of defective genes, inclusive of *ABL1*, *AXL*, *BRAF*, *CDC25A*, *CDKN2A*, *IGF1R*, *KRAS*, *MECOM*, *MMP1*, *MYC*, *NOTCH1*, *NRAS*, *PIK3CG*, *PTK2*, *RPTOR*, *SPTBN1*, *STAT2*, *TNKS* and *ZHX2*, as possible candidates for roles in chemosensitivity for compound MOAs that target primarily, but not exclusively, kinases, nucleic acid synthesis, protein synthesis, apoptosis and tubulin. These results find exploitable instances of enhanced chemosensitivity of compound MOA's for selected defective genes. Collectively these findings will advance the interpretation of pre-clinical screening data as well as contribute towards the goals of cancer drug discovery, development decision making, and explanation of drug mechanisms.

## Introduction

The emergence of extensive human tumor cell line compound screening data, coupled with advances in cancer genomic technologies, has generated comprehensive and complex databases [1]. Strategies for analyzing this data may identify important links between genetic changes that contribute to the hallmarks of cancer biology [2] and the discovery of leads in the pursuit of small-molecule cancer therapy [3]. The present report examines links between genetically defective genes in the National Cancer Institute's panel of sixty tumor cell lines (referred to hereafter as NCI60), chemosensitivity, as measured by growth inhibition ($\underline{GI50}_{NCI60}$; adopting the convention of an under bar to describe the vector of $\underline{GI50}_{NCI60}$ (N = 59) measurements for each screened compound) and preferences for mechanisms of action (MOA) of identified linkages. An elegant study by Ikediobi et al. [4] addressed this goal by examining relationships between mutations in 24 cancer genes in the NCI60 tumor cell

Contracts No. HHSN261200800001E and
HHSN261201700007I. There was no additional
external funding received for this study.

**Competing interests:** No competing interests.

**Abbreviations:** NCI, National Cancer Institute; FDA,
Food and Drug Administration; GI50, tumor cell
Growth Inhibition; SOM, Self Organizing Map; DTP,
Developmental Therapeutics Program; MOA,
Mechanism of Action; NSC, Cancer Chemotherapy
National Service Center; MUT, gene mutation; CN,
copy number alteration; FUSION, gene fusion or
splice; Tu, tubulin; T1, topoisomerase I; T2,
topoisomerase II; A, alkylating agent; D, DNA
interacting agent; PK, protein kinase; Apo,
apoptosis; Ho, hormone; HDAC, histone
deacetylase; HSP90, heat shock protein 90; PSM,
proteasome; $GI50_{codebook}$, vector of tumor cell
responses for each SOM node; $GI50_{component}$, SOM
response for each tumor cell.

lines and the $GI50_{NCI60}$ activity of ~8k screened compounds. Their finding of a strong associa-
tion between the *BRAF* mutation (V600E) and the $GI50_{NCI60}$ activity of phenothiazines sup-
ports important links between altered genes, chemosensitivity and MOAs. The current
analysis extends this work, with significant differences.

- $GI50_{NCI60}$ results for ~53k screened compounds are analyzed (DTP database).

- A larger set of gene mutations (N = 368) for the NCI60 tumor cell lines are analyzed (CBio-
Portal database).

- A novel analysis of $GI50_{NCI60}$, based on Self-Organizing-Maps (SOMs), emphasizing FDA
approved compounds with assigned MOAs in the NCI60 screened compounds, is used to
derive links between tumor cell chemosensitivity, genetically altered genes and MOAs.

Efforts to develop links between pre-clinical tumor cell screening data, genomic defects and
drug mechanisms may contribute to advances in small-molecule cancer therapies. An impor-
tant element of these efforts requires more informed interpretations of small-molecule screen-
ing results in the context of genomic profiles and drug action. These associations may yield
undiscovered opportunities for drug re-purposing and new applications of gene mutations
towards personalized medicine.

## Data

Three publicly available data sources are used for this analysis. First, chemosensitivity data
consists of the 2019 release of $GI50_{NCI60}$ measurements from the Developmental Therapeutics
Program (DTP) in the National Cancer Institute. Historically the NCI60 screen was designed
to identify relationships between chemotypes and cellular responses [5]. Their bulk download
(https://dtp.cancer.gov/discovery_development/nci-60) includes GI50 values for 159 tumor
cell types. A subset of 70 tumor cell lines, identified previously [6] as representing an informa-
tion-rich component of this data, consists of ~53k screened compounds, which for this analy-
sis was reduced to 46,798 $GI50_{NCI60}$ records when filtered for a coefficient of variation above
0.1. Z-score normalized $GI50_{NCI60}$ measurements of the filtered data (~46K) were used for
chemosensitivity analysis. The raw data file is included in the **S1 master_appendix sheet
GI50**.

Second, genetic data is obtained from the cBioPortal database (https://www.cbioportal.org/
) [7,8]. A total of 368 altered genes are listed for the NCI60; with either a mutation (MUT),
copy number alteration (CNA) or fusion/splice (FUSION). These genomic changes are
grouped, so that a gene alteration due to any or all types of variations will be designated as
genetically defective. In this context, a defective gene indicates only a modification from the
wild-type allele. Genes designated as defective genes can have wide-ranging effects including
gain-of and/or loss-of gene function. Defective genes occur within each NCI60 tumor cell indi-
vidually or as pairs, doublets, triplets, etc. **S1 Appendix Fig 1** in S1 File displays a histogram
for the frequency of defective genes within the NCI60. The highest frequency exists for tumor
cell lines having a single defective gene. This frequency decreases progressively down to less
than one percent for tumor cell lines sharing 10 defective genes. The cumulative frequency of
tumor cell lines sharing any defective gene is 0.97, an indication that the probability of tumor
cell lines sharing any defective gene is approaches one. **S2 Appendix Fig 2** in S1 File displays
the histogram of defective genes shared between tumor cell lines. These results find that shared
defective genes, comprised of doublets and triplets are more common compared to the appear-
ance of only a single defective gene (consistent with Ikediobi et al. [4]). **S3 master_appendix
sheet appendix_table_I** lists the singlets, doublets and triplets of defective genes observed in

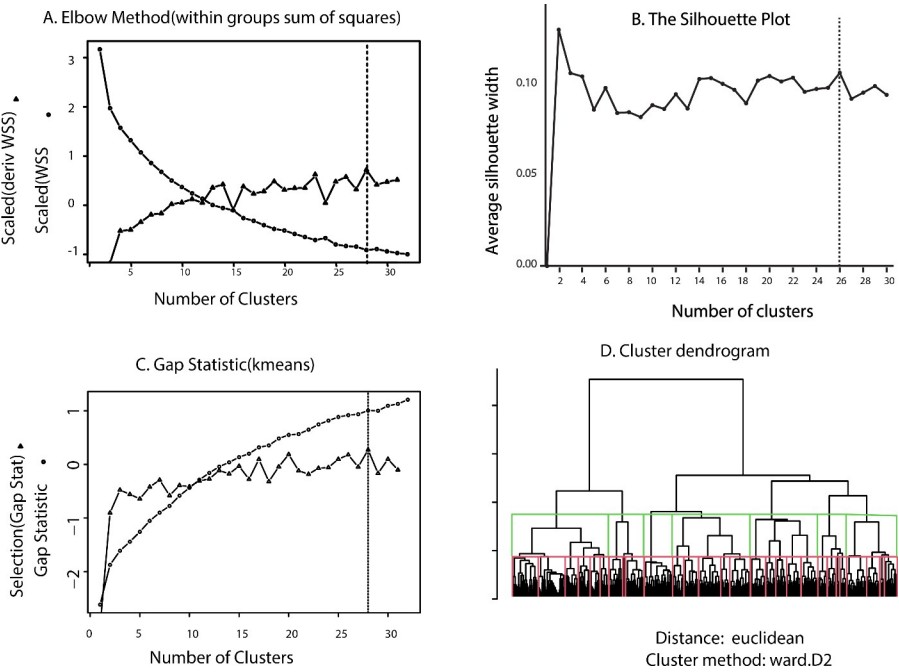

**Fig 1. Results for selecting optimal cluster size using the elbow method (Panel A), silhouette method (Panel B) and the gap_statistic method (Panel C).** Selection for the **elbow** method is based on the largest local derivative of the within groups sum of squares (**Panel A** triangles), the maximum silhouette width (**Panel B**) and the first non-negative value for Gap(k)-(Gap(k + 1)−sd$_{k+1}$) (**Panel C** triangles). These results indicate an optimal number of clusters in the 26–28 range. **Panel D** displays the GI50$_{codebook}$ dendrogram (Euclidean, Ward's) with cuts at 28 (red lines) and 7 clusters (green lines), respectively.

the NCI60. **S4 master_appendix sheet appendix_table_II** summarizes their counts. Inspection finds a diverse set of defective genes, some of which are not considered to have important roles in cancer. *CDC25A*, *TP53*, *CDKN2A*, *CDKN2B*, *MYC*, *BRAF*, *EP300*, *KRAS*, *NOTCH1* and *PTK2* are the top ten most frequently occurring defective genes. To summarize, defective genes appearing as doublets or triplets finds these top ten defective genes to appear in combination with themselves and other genes. Collectively these results indicate that shared defective genes, with diverse roles in cellular biology, are common within the NCI60.

Third, CellMiner [9] (https://discover.nci.nih.gov/cellminer/home.do) provides information about mechanism of action (MOA) for Food and Drug Administration (FDA) approved compounds. CellMiner reports 270 FDA compounds with unique NSC (National Service Center) and Name assignments that have been screened in the NCI60 (ca. 2019). One-hundred and ninety FDA screened compounds appear in the 46,798 GI50$_{NCI60}$ responses. One-hundred and four MOA assignments exist for this set of compounds. These assignments consist of a primary MOA designation followed by secondary MOAs. For example the assignment BCR-ABL|YK,FYN,LYN indicates BCR-ABL at the primary MOA, with YK (tyrosine kinase), FYN and LYN (both Proto-Oncogene, Src Family Tyrosine Kinases) as secondary MOAs. Thirty primary MOAs are assigned to this data. The complete set of MOAs for FDA screened compounds is listed in **S1 S5 master_appendix sheet appendix_table_III**. Seven of the most frequent primary MOA classes spanning this data function to target tubulin:**Tu**, topoisomerase 2:**T2**, topoisomerase 1:**T1**, alkylation:**A** (A2: Alkylating_at_N-2_position_of_guanine, A6: Alkylating_at_O-6_of_guanine, A7: Alkylating_at_N-7_position_of_guanine, AlkAg: Alkylating agent and anti-metabolites:**AM**), DNA:**D** (Db:DNA_binder, DDI/R,

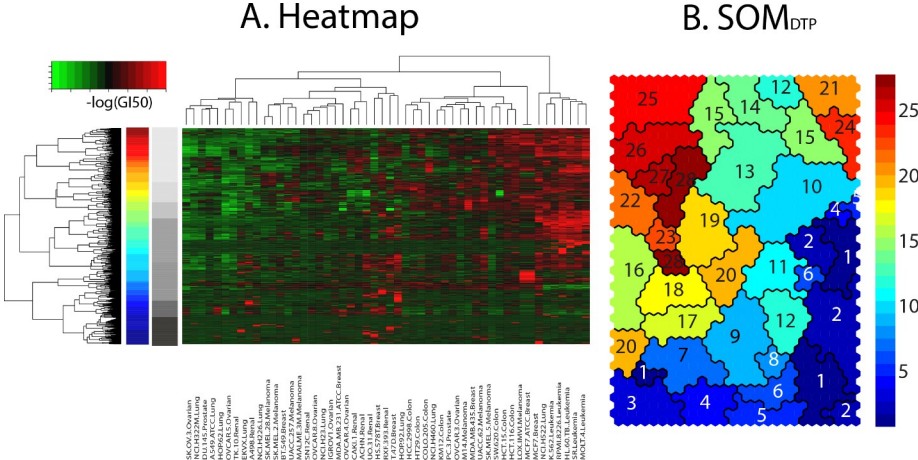

**Fig 2. Panel A displays a heatmap of <u>GI50</u><sub>codebook</sub>, colored spectrally from green(chemoinsensitive) to red (chemosensitive) response.** Dendrogram at the left represents hierarchical clustering (Euclidean, Ward's) of <u>GI50</u><sub>codebooks</sub> (reproduced from **Fig 1 Panel D**). **Panel B** displays the SOM<sub>DTP</sub> colored according to hierarchical cutree [18] specified at the optimal number of 28 meta-clades. The 28 colors appear spectrally from meta-clade 1 (dark blue), at the bottom of the hierarchical dendrogram, to meta-clade 28 (dark red), at the top of the hierarchical dendrogram. Grayscale bar adjacent to the 28 meta-clade spectrally colored bar displays the 7 meta-clades groupings. The NCI60 tumor cell lines clustered in the heatmap are ordered, left to right, as: SK.OV.3.Ovarian, NCI.H322M.Lung, DU.145. Prostate, A549.ATCC.Lung, HOP.62.Lung, OVCAR.5.Ovarian, TK.10.Renal, EKVX.Lung, A498.Renal, NCI.H226. Lung, SK.MEL.28.Melanoma, SK.MEL.2.Melanoma, BT.549.Breast, UACC.257.Melanoma, MALME.3M.Melanoma, SN12C.Renal, OVCAR.8.Ovarian, NCI.H23.Lung, IGROV1.Ovarian, MDA.MB.231.ATCC.Breast, OVCAR.4.Ovarian, CAKI.1.Renal, ACHN.Renal, UO.31.Renal, HS.578T.Breast, RXF.393.Renal, T.47D.Breast, HOP.92.Lung, HCC.2998. Colon, HT29.Colon, COLO.205.Colon, NCI.H460.Lung, KM12.Colon, PC.3.Prostate, OVCAR.3.Ovarian, M14. Melanoma, MDA.MB.435.Breast, UACC.62.Melanoma, SK.MEL.5.Melanoma, SW.620.Colon, HCT.15.Colon, HCT.116.Colon, LOX.IMVI.Melanoma, MCF7.ATCC.Breast, MCF7.Breast, NCI.H522.Lung, K.562.Leukemia, RPMI.8226.Leukemia, HL.60.TB..Leukemia, SR.Leukemia, MOLT.4.Leukemia, CCRF.CEM.Leukemia.

DNA_damage_repair/inducer, Df:antifols, Ds: DNA_synthesis_inhibitor), kinases:**PK** and apoptosis:**Apo**. MOA:**PK** consists of over 100 kinase targets. FDA compounds screened in the NCI60, and their assigned CellMiner MOA, will be used for the linking MOA to chemosensitivity.

## METHODS: Data clustering

The methods for linking chemosensitivity, defective genes and MOA apply a sequential, multi-tiered approach. First, the <u>GI50</u><sub>NCI60</sub> data is organized into clusters. Many statistical tools are now available for clustering <u>GI50</u><sub>NCI60</sub> data [10]. Relying on our prior analysis [6], the results presented here use Self-Organizing-Maps (SOMs) [11,12]. Parameters from prior SOM analyses are selected for clustering (hexagonal nearest neighbors, Epanechnikov Function kernal [13]). SOM dimensions are based on a heuristic using the ratio of the first and second principal components of the data. The y-axis dimension is calculated as round(sqrt(munits/ratio*sqrt (0.75))), where munits = 5*nsamples^0.543. The x-axis dimension is calculated as round (munits/y-dimension). This heuristic is derived from the developers of SomPak, based on their usage. The sqrt(0.75) multiplier is explained as follows, "in the hexagonal lattice, the side lengths are not directly proportional to munits (= 5*nsamples^0.543) since the units on the y-axis are squeezed together by a factor of sqrt(0.75)". Applying this procedure yields SOM map dimensions of 44 rows and 28 columns.

Each of these 1232 SOM nodes defines a vector representing the average <u>GI50</u><sub>NCI60</sub> for all compounds clustered within each SOM<sub>DTP</sub> node (referred to hereafter as a node's

GI50$_{codebook}$). **S2 master_appendix sheet SOM_codebook** lists the 1232 GI50codebooks. Each compound's SOM$_{DTP}$ node will be referred to as its projection. The best projection can be extended to include the 2$^{nd}$, 3$^{rd}$, 4$^{th}$, etc. SOM$_{DTP}$ nodes to determine whether a compound's next best projections appear as SOM$_{DTP}$ neighbors. Prior analyses found GI50$_{codebook}$ patterns to be associated with a compound's MOA (e.g. alkylating agents, tubulin targeting agents, DNA/RNA damaging agents and agents affecting mitochondrial function [6]). Analysis of GI50$_{codebook}$ patterns has also been proposed for use in the development of clinical strategies based on differentially expressed molecular targets within classes of tumors [14,15]. Other applications include the recent identification of unique GI50$_{codebook}$ patterns within the NCI60 renal subpanel as the basis for further testing of the natural product-derived family of englerins [16].

While each SOM$_{DTP}$ node represents a cluster of GI50$_{NCI60}$ values, a more global, lower resolution representation, that optimally groups SOM$_{DTP}$ nodes into meta-clades, is proposed. Three state-of-the art procedures are used to determine the optimal number of meta-clades; the **elbow** method minimizes the within-cluster sum of squares, WSS (a measure of within cluster similarity) and maximizes the between-cluster sum of squares (a measure of how separated each cluster is from the others), the **silhouette** method computes the average silhouette of observations for different numbers of clusters (selecting the optimal cluster size that maximizes the average silhouette) and the **gap_statistic** method [17] determines the total within intra-cluster variation for different numbers of clusters (selecting the cluster size that maximizes the gap_statistic). **Fig 1** displays the results for these three methods applied to GI50$_{codebooks}$. The optimal cluster size is indicated by the vertical lines in each plot. The **elbow** and **gap** methods rely on an inflection point on each curve for optimal cluster size, while the **silhouette** method seeks the largest value for silhouette width. **Panel A** displays WSS as circles for the **elbow** method and the first derivative of WSS, normalized by the local average WSS, as triangles. The maximum value of the derivative of WSS occurs for 28 clades. The **silhouette** method yields a maximum value at 26 clusters (cf. **Panel B**). The criterion for optimal cluster size using the **gap** statistic seeks the smallest number of clusters such that the gap statistic is within one standard deviation of the next gap statistic: Gap(k)≥Gap(k + 1)−sd$_{(k+1)}$ (displayed as triangles in **Panel C**), yields 28 clusters as optimal. **Panel D** displays the GI50$_{codebook}$ cluster dendrogram using the cutree tool [18] to group the dendrogram into 28 meta-clades (red lines) and 7 meta-clades (green lines). Based on these results a value of 28 was selected for the optimal number of meta-clades used in this analysis. The rationale for cutting the dendrogram at 7 clusters will be provided later in the analysis of MOAs.

A visual perspective of SOM$_{DTP}$ and the 28 meta-clades appears in **Fig 2**. The 1232 GI50$_{codebooks}$ appear as a clustered heatmap (Euclidean,Ward's) in **Panel A**. The dendrogram at the left edge of the heatmap, displays the dendrogram appearing in **Fig 1, Panel D**. The vertical ribbon, adjacent to this dendrogram, colored spectrally from blue to red, represents the subdivision of the hierarchal clade tree into 28 meta-clades. The pvclust utility [19], using random resampling, confirms this set of 28 meta-clades with a confidence p-value above 0.995 across resampling (n = 1000 resamples). **Panel B** in **Fig 2** displays the 28 meta-clades on SOM$_{DTP}$, colored according to the spectral-colored vertical ribbon in the left panel. The data reduction of 1232 SOM$_{DTP}$ nodes to 28 meta-clades yields a lower resolution, more manageable, perspective of the complete SOM$_{DTP}$. The gray ribbon in **Panel B** displays the dendrogram cut into 7 major groups. Groupings consist of **A**: meta-clades 1–6, **B**: meta-clades 7–9, **C**: meta-clades 10–15, **D**: meta-clades 16–18, **E**: meta-clades 19–20, **F**: meta-clades 21–24 and **G**: meta-clades 25–28. The vertical grayscale colored bar in **Fig 2** displays these seven groupings from **A**(bottom:black) to **G**(top:light gray). SOM$_{DTP}$ meta-clades will be assessed according to the differential chemosensitivity of tumor cell lines with and without defective genes.

Noteworthy is the mapping of the 28 cutree clades to discontinuous $SOM_{DTP}$ regions. Ideally, cutree clades might appear as contiguous regions the 2-dimensional $SOM_{DTP}$. However, this is not the case. To obtain contiguous $SOM_{DTP}$ regions, an alternative hierarchical clustering algorithm would need to be used that only combines adjacent dendrogram clades that appear beside each other on $SOM_{DTP}$. Although not pursued here, assigning contiguous SOM regions is an active area of research in dimensionality reduction [20], with specific focus on representing SOMs in one dimension [21]. Many of these efforts use randomized resampling to identify contiguous map regions by consensus. Usually standard hierarchical clustering suffices, and any outlying (noncontiguous) points can be accounted for manually. Towards this end, $SOM_{DTP}$ singletons, appearing as a hierarchical clade that maps to $SOM_{DTP}$ as a node without the same meta-clade neighbors, have been replaced by their neighborhood meta-clade assignments. There are 12 such cases (0.0097 = 12/1232) for this data set.

An additional consideration for non-contiguous $SOM_{DTP}$ meta-clades may result from the assignment of distances used for clustering (Euclidean for hierarchical clustering and Epanechnikov Function [13] for SOMs). Our choice of the Epanechnikov Function for SOM clustering consistently yielded the lowest $SOM_{DTP}$ quantization errors [6]. However, a more likely explanation for non-contiguous $SOM_{DTP}$ meta-clades involves differences in clustering methodology. SOMs organize data by mapping each cluster to its most similar neighbors (six in the case of hexagonal mapping); whereas hierarchical clustering, as used to obtain heatmap dendrograms, builds each branch of the dendrogram by pairwise associations. The failure to map hierarchical clustering methods directly to contiguous $SOM_{DTP}$ regions is not unexpected and points more to the limitations of hierarchical methods to match non-hierarchical methods, regardless of distance metrics.

## METHODS: Identification of $SOM_{DTP}$ nodes with enhanced chemosensitivity

$SOM_{DTP}$ nodes are analyzed for enhanced chemosensitivity of tumor cell lines with versus without defective genes. Each $GI50_{codebook}$ is divided into subsets comprising tumor cell lines with ($GI50_{defective}$) and without ($GI50_{wild-type}$) a defective gene. A Student's t-test is used to identify cases of relatively higher chemosensitivity for $GI50_{defective}$ versus $GI50_{wild-type}$. $SOM_{DTP}$ nodes with Student's p-values less than or equal to 0.05 were further assessed for statistical significance by bootstrap resampling [22,23]. Each node's $GI50_{codebook}$ was randomly shuffled and a Student's t-test performed, while maintaining the tumor cell's wild-type and defective gene status. One-thousand trials were conducted for each $GI50_{codebook}$ and a p-value was estimated by counting the number of times the shuffled p-value was less than the initial, unshuffled, p-value. Dividing this value by 1000 yields an estimate for the probability of the observed p-value occurring by chance. $SOM_{DTP}$ nodes with measured p-values less than 0.05 and below their estimated chance occurrence were accepted for further analysis. Sixty-five percent (65%, n = 635) of the 1232 SOM nodes pass this criterion and account for 121 defective genes.

**Fig 3** summarizes the results for $GI50_{codebook}$ at $SOM_{1,13}$ (subscripting refers to the SOM node, i.e. $SOM_{row,column}$). Five NCI60 tumor cell lines have the defective *ABL1* gene; with these tumor cell lines having a mean $GI50_{defective}$ response nine-fold higher than $GI50_{wild-type}$ (p = 6.91e-3). **Panel A** in **Fig 3** displays $GI50_{codebook}$, ordered from most chemosensitive to least chemosensitive values. NCI60 tumor cell lines with the *ABL1* alteration, highlighted in red and representing $GI50_{defective}$, are ranked at positions 3, 6, 8, 25 and 51. $SOM_{DTP}$ can also be viewed according to each NCI60 tumor cell (referred to as $GI50_{component}$). **Panel B** of **Fig 3** displays $GI50_{component}$ for each of the 5 tumor cell lines with defective *ABL1*. Regions of greatest and least chemosensitivity for each tumor cell are displayed spectrally from red to blue,

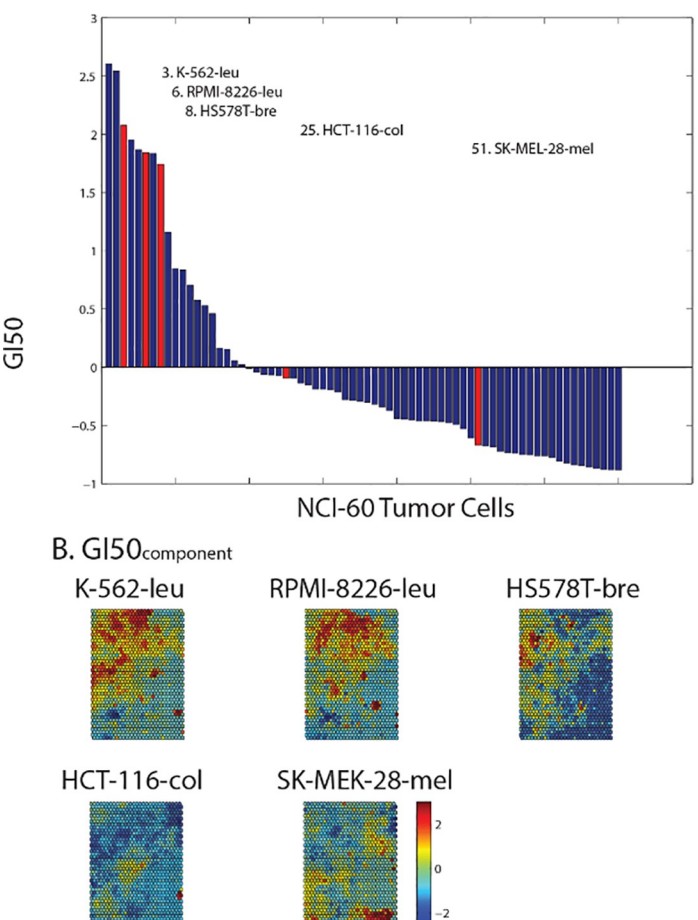

**Fig 3. Panel A displays GI50$_{codebook}$ for SOM$_{1,13}$, ordered from most to least chemosensitivity.** The 5 tumor cell lines with the defective *ABL1* gene appear as red bars. **Panel B** displays GI50$_{component}$ for the 5 tumor cell lines with defective *ABL1*. SOM$_{DTP}$ nodes are colored spectrally from highest chemosensitivity (red) to lowest chemosensitivity (blue).

respectively. Noteworthy is the location of Gleevec chemosensitivity to the most sensitive (e.g. red) GI50$_{component}$ SOM$_{DTP}$ regions for K-562, RMPI-8226 and HS578T.

Panel A of **Fig 4** projects onto the SOM$_{DTP}$ the Students t-statistic for tumor cell lines with defective *ABL1*; where the t-statistic values are colored spectrally from low(blue) to high(red) significance. SOM$_{DTP}$ nodes without statistical significance (p>0.05) are not colored. The most significant t-statistics for defective *ABL1* are located mainly in SOM meta-clades 1, 10, 14 and 26. Gleevec appears as the most significant SOM$_{5,15}$ node in meta-clade 14. For comparison, the results for *KRAS* are projected in **Panel B** of **Fig 4**. There are 12 tumor cell lines (A549/ATCC-Lung, CCRF-CEM-Leukemia, HCC-2998-Colon, HCT-116-Colon, HCT-15-Colon, HOP-62-Lung, NCI-H23-Lung, NCI-H460-Lung, OVCAR-5-Ovarian, RPMI-8226-Leukemia, SK-OV-3-Ovarian and SW-620-Colon) harboring defective *KRAS*, with significant chemosensitive SOM$_{DTP}$ nodes appearing in meta-clades 21, 22 and 27. SOM meta-clade 21 is the location of the FDA compound cytarabine (ara-C) and is consistent with the conclusion of Ahmad et al [24] that adult AML patients carrying defective *KRAS* benefit from higher ara-C doses more than wt *KRAS* patients.

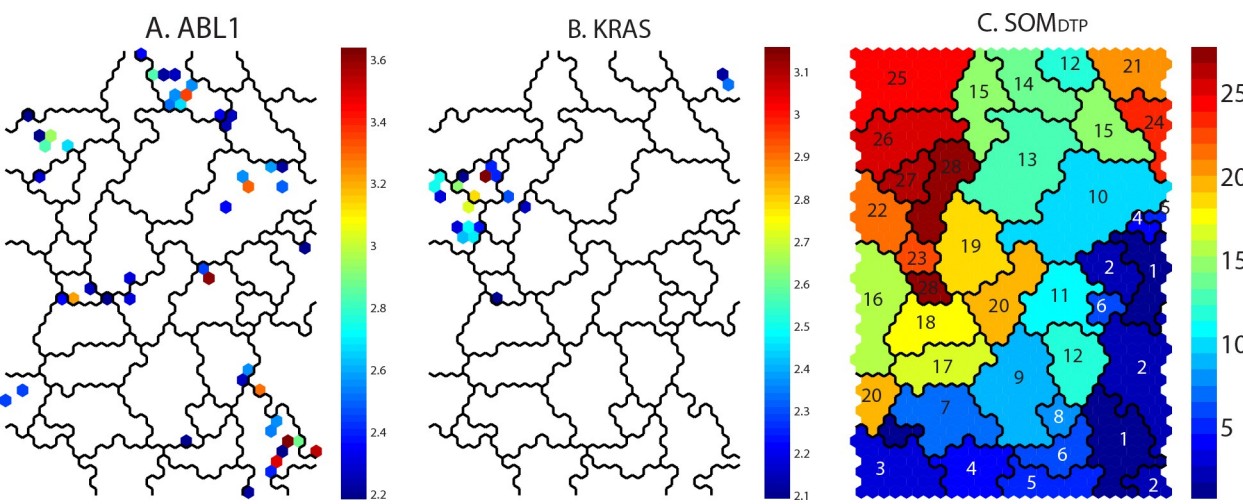

**Fig 4. Panels A and B display significant chemosensitive SOM$_{DTP}$ nodes (projected as their t-statistic from a Student's t-test; blue:least, red: most significant) for tumor cell lines with defective *ABL1* and *KRAS*, respectively. Panel C** displays the 28 SOM$_{DTP}$ meta-clades.

**S3 and S4 Appendix Figures 3** and **4** in S1 File display additional examples for the defective genes *PIK3RI* and *IGF1R*, respectively. *PIK3R1* (Phosphatidylinositol 3-Kinase Regulatory Subunit Alpha) and a related gene, *PIK3CA* (PI3-Kinase Subunit Alpha) are lipid kinases capable of phosphorylating the 3'OH of the inositol ring of phosphoinositides. Both are responsible for coordinating a diverse range of cell functions including proliferation and survival. Defective *PIK3CA* has been documented by Whyte and Holbeck [25] to enhance tamoxifen sensitivity in selected NCI60 tumor cell lines. The results here also find chemosensitivity in NCI60 tumor cell lines having defective *PIK3R1*. The second example of defective *IGF1R* supports the importance of evaluating drug sensitivity for compounds targeting leukemia cell lines [26] and the emergence of *IGF1R* as a potential therapeutic target for the treatment of different types of cancer including plasma cell myeloma, leukemia, and lymphoma [27]. Both examples illustrate potential role of defective genes in chemosensitivity.

The Students t-statistic represents the significance when comparing the chemosensitivity of a SOM$_{DTP}$ node for tumor cell lines with, versus without, defective genes. Parametric tests, such as the Student's t-test, are applicable over non-parametric tests (Wilcoxon/Whitney/ Mann, Kruskal-Wallis) when the underlying sample distribution is known and normal. The data analyzed here represents a strongly normal distribution (p < 0.001, lognormal test) with small deviations at the tails from a linear log normal quantile-quantile plot; supporting the use of a parametric statistic.

The application of a bootstrap procedure to cases with a significant Student's t-test is applied as a correction against Type I error for the following reasons. First, a bootstrap method can be used to estimate the sampling distribution of GI50$_{codebook}$ for each SOM$_{DTP}$ node. This test utilizes the node's codebook vector as the initial sample representative and applies a bootstrap procedure to estimate the sampling distribution. Since 1000 samples were used, the p-value estimate corrects the empirical estimate using a division by 1000. This correction parallels multiple test corrections for traditional statistics [28]. Second, in this design there are 1000 statistical tests performed for each of the 1232 SOM$_{DTP}$ nodes. An important caveat of jointly using GI50$_{codebook}$ to create some type of correction for multiple test is their lack of independence [11,12]. This non-independence is disallowed when applying a Bonferroni, Holm or Benjamini-Hochberg [29] corrections. None-the-less, the long-standing debate continues to

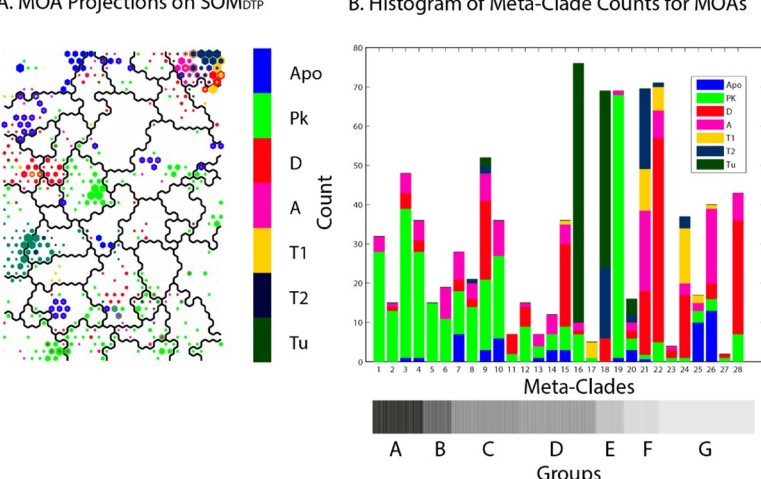

**Fig 5. Panel A: SOM$_{DTP}$ projections for FDA approved compounds for the primary CellMiner assigned MOAs.**
Projections include the top 10$^{th}$ percentile of SOM$_{DTP}$ nodes for each compound. **Panel B:** histogram of the counts for these primary MOAs across SOM meta-clade groups. Primary MOAs appear color-coded in each vertical bar, with their heights corresponding to MOA counts in each meta-clade. Horizontal grayscale bar below Panel B indicates meta-clade groups **A:G** (reproduced from **Fig 2** Panel A).

exist about bootstrap applications, possible overestimation of 'true' values, and appropriate corrections for random noise [30].

## Methods: Mapping MOA to SOM$_{DTP}$

SOM$_{DTP}$ projections for the most frequent primary CellMiner MOA assignments (Tu, T1, T2, A, D, Apo and PK) are displayed **Panel A** of **Fig 5**. **Panel B** displays the histogram of SOM$_{DTP}$ node counts for these MOA assignments. Inspection indicates that MOA classes A, D, T1 and T2 appear mainly in the upper right SOM$_{DTP}$ region (SOM meta-clade 21; Group **A**), while MOA Apo appears mainly in the upper left region (SOM$_{DTP}$ meta-clades 25 and 26; Group **G**). Tu compounds are found mainly in SOM$_{DTP}$ meta-clades 16, 17 and 18 (Group **D**). SOM$_{DTP}$ meta-clade 19(Group **E**) consists of only MOA PK; while MOA PK compounds are in the majority for SOM$_{DTP}$ meta-clades 1 through 6(Group **A**). The horizontal gray scale bar at the bottom of the right panel identifies the seven meta-clade groups assigned earlier (cf. **Figs 1** and **2**). Inspection indicates relative similarities of MOA types within each of the seven meta-clade groups **A:G**. Notable is the majority representation of MOA:PK in Group **A** and MOA:Tu in Group **D**. Detailed results for MOAs across meta-clade groups will be presented later.

**Table 1** lists the most frequent primary MOAs for meta-clade groups **A:G**. These results segregate the primary MOAs into separate regions of the cutree = 7 hierarchical dendrogram

**Table 1. Most frequent primary MOA assignments within meta-clade groups A:G.**

| Meta-clade Groups | MOA |
| --- | --- |
| A: 1–6 | PK |
| B: 7–9 | PK, A |
| C: 10–15 | PK, A, D |
| D: 16–18 | Tu, T2 |
| E: 19–20 | PK |
| F: 21–24 | A, D, T1 |
| G: 25–28 | Apo, A, D |

(gray bar in **Fig 2** **Panel A** and **Fig 5** **Panel B**); MOA:PK appears at the bottom, MOAs targeting DNA and Apo appear at the top and mixtures of primary MOAs appear in the middle. These meta-clade grouping will be analyzed in greater detail for links of MOAs to defective genes.

## Methods: Enrichment of MOA

Fisher's exact and chi-square tests are used to identify cases where the $SOM_{DTP}$ projections of defective genes are statistically enriched in co-projections of MOA types. These tests are useful for categorical data that result from classifying objects in two different ways; and are used to determine a statistical measure for the random likelihood of the intersection of each classification. For each defective gene the number of $SOM_{DTP}$ nodes with significant Student's t-statistics are determined ($N_{gene}$). FDA approved compounds that co-project to $N_{gene}$ determine a unique set of MOA's associated with each defective gene ($MOA_{gene}$). All FDA compounds that share any $MOA_{gene}$ are collected ($N_{FDA}$); where the $10^{th}$ best FDA projections are included in the count. Extending FDA projections beyond only the best node achieves two goals. First, it establishes $SOM_{DTP}$ regions rather than individual nodes for MOA classification. Second, increasing the numbers in the contingency table extends significance testing to include Fisher's exact and the chi-square testing. The contingency table entries become; $p_{11}$ = intersection ($N_{gene},N_{FDA}$), $p_{12} = N_{gene}-p11$, $p_{21} = NFDA-p_{22}$ and $p_{22} = p_{11}$ by default to conserve equal row and column sums.

**Fig 6** illustrates a sample result of the steps for calculating the Fisher's exact statistic using *ABL1*. **Panel A** (reproduced from **Fig 4**) displays the significant $SOM_{DTP}$ nodes for defective *ABL1*, where $N_{gene} = 48$. Collecting the MOA's for the 11 FDA compounds co-projected to $N_{gene}$ finds 6 MOAs (Apo, Ho, HSP90, NonCan, PK and BCR-ABL). **Panel B** in **Fig 6** displays the top $10^{th}$ percentile of all FDA compounds sharing any one of these 6 MOAs, to yield $N_{FDA}$ = 189. Completing the contingency table with their intersection (22, results in a Fisher's exact score of 1.958262e-09 (logpval(-20.051208)).

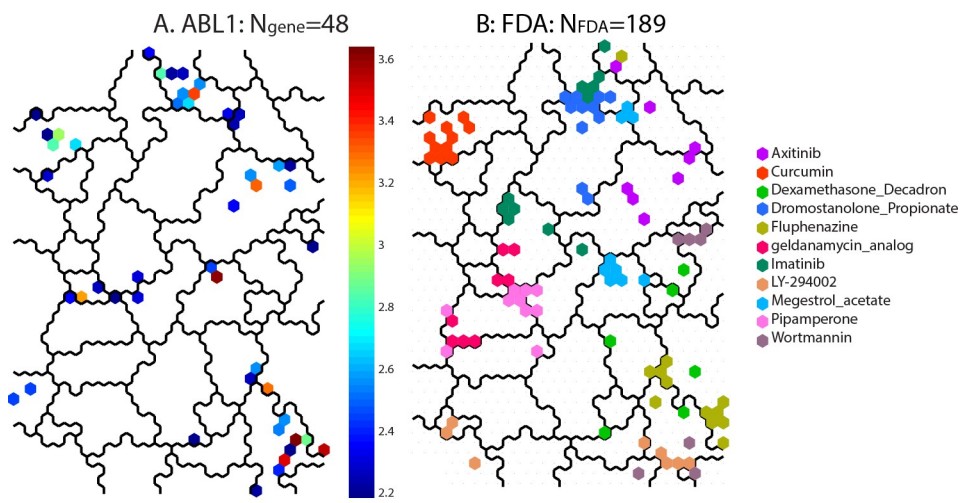

$N_{gene-FDA}=22$   p-value$_{fishers}$=1.96e-9

**Fig 6. Panel A displays the significant $SOM_{DTP}$ nodes for *ABL1* ($N_{gene}$ = 48).** Eleven FDA compounds are co-projected to $N_{gene}$; yielding 6 MOAs. The $SOM_{DTP}$ in **Panel B** displays the top $10^{th}$ percentile of projections for FDA compounds sharing these MOAs ($N_{FDA}$ = 189). The intersection of $N_{gene}$ and $N_{FDA}$ = 22, yielding a Fishers exact p-value of 1.958262e-09, log(p-value = -20.05).

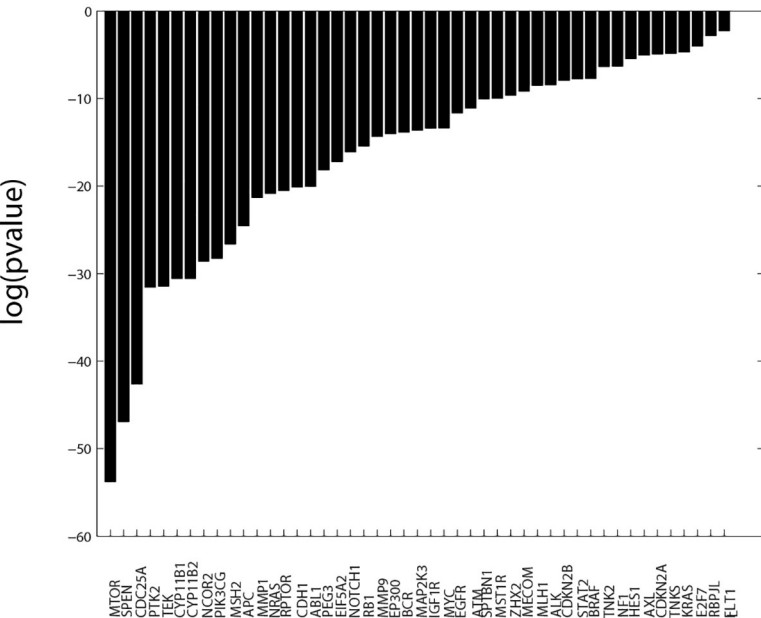

**Fig 7. Fisher's exact scores (log(pvalue), pvalue$< = 0.05$).** Results are based on classifications using up to the 10th best SOM projection nodes for FDA compounds. Forty-seven defective genes have significant Fisher's exact scores when tested over the complete SOM$_{DTP}$.

Fig 7 displays a bar chart of the log(pvalue) for the 47 genes with significant ($p< = 0.05$) Fisher's exact scores, when tested over the complete SOM$_{DTP}$. Defective genes with the top-most significance scores include *MTOR*, *SPEN*, *CDC25A*, *PTK2*, *TEK*, *CYP11B1*, *CYP11B2*, *NCOR2*, *PIK3CG*, *MSH2*, *APC* and *MMP1*. Note that this subset of defective genes is not exclusively associated with human cancers. Fisher's exact and chi-square tests for MOAs will be applied to meta-clade groups (**A-E**). The average log(pvalue) for both statistics will be reported as a contingency score.

## Results

The multi-tiered approach described in the **METHODS** builds a framework to achieve the study's goal of associating chemosensitivity, defective genes and MOA. In summary: chemo-sensitivity data($n = 46k$, GI50$_{NCI60}$) is clustered as SOM$_{DTP}$ ($n = 1232$, GI50$_{codebooks}$), subdi-vided into meta-clades ($n = 28$) and 7 meta-clade groups (**A-G**). Defective genes ($n = 368$) are analyzed according to significant chemosensitivity on SOM$_{DTP}$ ($n = 121$, Student's t/bootstrap) and enrichment for type of MOA of defective genes ($n = 47$ genes, contingency score; reported as the average log(Fisher's exact and chi-square scores). Contingency scores will be used to identify significant MOA enrichments for defective genes across meta-clade groups (**A-G**). The results for SOM$_{DTP}$ clustering will be presented first, followed by the results for MOA enrichments in groups **A-G**.

### Results: SOM$_{DTP}$

Fig 8 **Panel A** displays SOM$_{DTP}$, colored according to similarity of neighboring GI50$_{codebooks}$; where the most similar GI50$_{codebook}$ neighbors are displayed in deep red and the most dis-sim-ilar GI50$_{codebook}$ neighbors appear in bright yellow. The 28 optimal meta-clade boundaries are displayed as a black line, with the boundaries of the 7 meta-clade groups super-imposed as a

A. SOM~DTP~ projections for CellMiner derived compounds projected on 44x28 SOM nodes for ~46k screened compounds

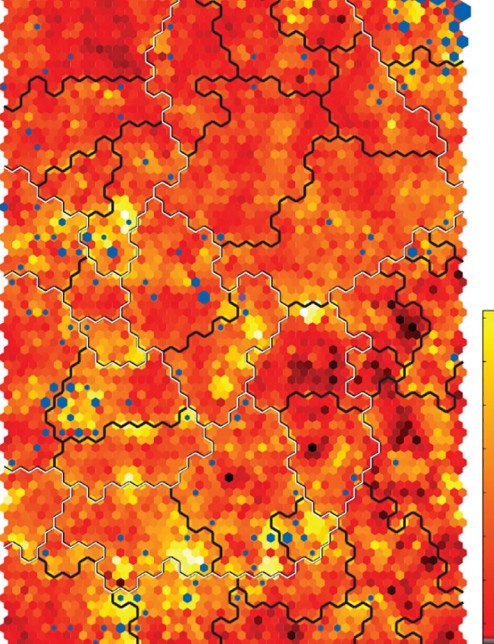

B. Histogram of Euclidean distances between GI50codebooks (top: with FDA compounds, bottom: without)

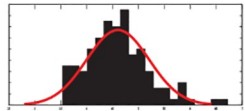

histogram of som inter-nodes distances for nodes containing the 270 cell miner nscs (mean_dist=4.43, n=127 nodes)

P(difference)=5.31e-7

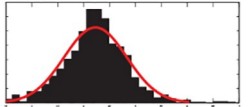

histogram of som inter-node distances for nodes not containing the 270 cell miner nscs(mean_dist=4.16, n=1105 nodes)

34 FDA approved compounds lacked sufficient gi50 values to pass initial filters

C. Listing of FDA compounds grouped by meta-clade.

| meta-clade | FDA Name |
|---|---|
| 1 | Gefitinib, Lapatinib |
| 3 | Dasatinib, Ibrutinib, Pazopanib |
| 5 | Cabozantinib, Staurosporine |
| 8 | Hydroxystaurosporine, Cordycepin, Decitabine, Depsipeptide, Midostaurin, Rapamycin, Temsirolimus |
| 9 | Int_to_TDP_665759, Acetalax, bisacodyl, Elesclomol, Everolimus, Fulvestrant, Mithramycin, Seliciclib, Simvastatin, SR16157 |
| 10 | Olaparib, Pazopanib, Rapamycin, Salinomycin |
| 13 | Dromostanolone_Propionate |
| 14 | Belinostat, Imatinib, Imexon, Megestrol_acetate |
| 15 | Arsenic_trioxide, Buthionine_sulphoximine, Fenretinide |
| 16 | Belinostat, Benzimate, Crizotinib, Docetaxel, Dolastatin_10, Eribulin_mesilate, Pipamperone, Vinblastine |
| 18 | Actinomycin_D, Alvespimycin, Carfilzomib, Elliptinium_Acetate, Geldanamycin_analog, Tyrothricin, Vinblastine, Vincristine |
| 19 | Bafetinib, Cobimetinib, Dabrafenib, Hypothemycin, Nilotinib, PD-98059, Pimozide, Selumetinib, Sunitinib, Trametinib, Vemurafenib, Vorinostat |
| 21 | 5-fluoro_deoxy_uridine_10mer, 7-Ethyl-10-hydroxycamptothecin, Batracylin, BEN, Bleomycin, BN-2629, Chlorambucil, Cisplatin, Cytarabine, Daunorubicin, Dexrazoxane, Doxorubicin, Epirubicin, Etoposide, Floxuridine, Gemcitabine, Hydroxyurea, Idarubicin, Irinotecan, Karenitecin, LMP-400, LMP776, Melphalan, Mitomycin, Mitoxantrone, Nitrogen_mustard, Pemetrexed, Pipobroman, Raltitrexed, RH1, Teniposide, tfdu, Thiotepa, Topotecan, Triethylenemelamine, Uracil_mustard, Valrubicin |
| 22 | CUDC-305, Fluorouracil, Methotrexate, Oxaliplatin, Palbociclib, Pelitrexol, Pralatrexate, Pyrazoloacridine |
| 24 | Digoxin, LOR-253, 3-Bromopyruvate |
| 25 | Lapachone, Lomustine, Curcumin |
| 26 | Dimethylaminoparthenolide, Obatoclax, Raloxifene |
| 28 | 6-Mercaptopurine, 8-Chloro-adenosine, AT-13387, Azacitidine, LDK-378, PX-316, Triciribine_phosphate |

D. SOM~DTP~ with FDA projections displayed as blue hexagons. Meta-clades regions are labelled by number.

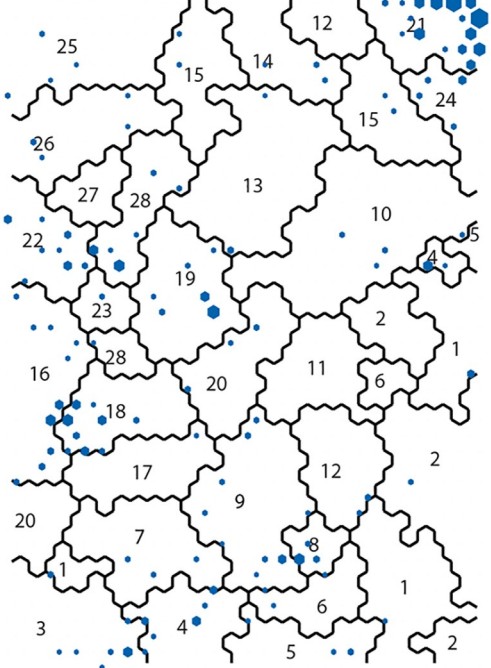

**Fig 8. Panel A.** SOM$_{DTP}$ is colored according to similarity of GI50$_{codebooks}$, where the most similar node neighbors are displayed in deep red and the most dis-similar node neighbors appear in bright yellow (see vertical bar adjacent to SOM$_{DTP}$). The 28 optimal meta-clade boundaries are displayed as a black line, with the boundaries of the 7 meta-clade groups super-imposed as a white line. FDA approved compounds are projected onto SOM$_{DTP}$ as blue hexagons, where hexagons are sized according to the number of FDA agents appearing in any node. **Panel B** displays the between node GI50$_{codebook}$ Euclidean distances for nodes with FDA compound projections (top) and without (bottom). **Panel C** lists FDA compound names grouped by 28 meta-clades. **Panel D** displays SOM$_{DTP}$ with FDA compounds (blue hexagons), meta-clade boundaries (solid lines) and meta-clade labels as numbers. FDA approved projections to SOM$_{DTP}$ nodes are listed in **S5 master_appendix sheet appendix_Table_III**.

white line. Two distinctive features characterize SOM$_{DTP}$. First, the best projections of FDA approved compounds appear as blue hexagons in **Fig 8 Panel A**, where hexagons are sized according to the number of FDA agents appearing in any SOM$_{DTP}$ node. Inspection finds a general tendency for approved agents to project to SOM$_{DTP}$ nodes with unique patterns (e.g. dissimilar GI50$_{codebooks}$). Statistical support for this observation is displayed in **Fig 8 Panel B,** in the form of histograms based on intra-node GI50$_{codebook}$ distances for nodes containing FDA approved agents (top histogram) and lacking FDA approved agents (bottom histogram). A Student's t-test for the vector distance between these two groups finds a p-value of 5.3e-7, in support of the visual association of FDA compounds and unique (e.g dis-similar) chemosensitivity patterns. Second, compound names for FDA screened agents are listed as a table in **Fig 8 Panel C** and projected on to SOM$_{DTP}$ in **Fig 8 Panel D**. A listing of these nodes and their SOM$_{DTP}$ projections also appears in **S5 master_appendix sheet appendix_Table_III**. In brief, FDA compounds with known MOAs are grouped together, with, for example, nucleic acid targeting agents appearing in the upper right corner of SOM$_{DTP}$ (meta-clade 21), tubulin targeting agents (meta-clades 16 and 18) and defective *BRAF* targeting agents (meta-clade 19). Collectively, these results support our prior report [6] of associations between NCI60 screened compounds, their MOAs and projections on SOM$_{DTP}$.

## Results: Group A (meta-clades 1 through 6)

The results for SOM meta-clade group **A** find twelve defective genes with significant contingency scores (*ABL1*, *ACVR2A*, *CDC25A*, *MMP1*, *MTOR*, *NCOR2*, *NF1*, *PIK3R1*, *RB1*, *RPTOR*, *SOX9* and *ZHX2*) associated with eleven MOA classes (PK, Ang, Ho, PARP, AM, BCR-ABL, NonCan, Db, HDAC, HYP and Pase). **Fig 9 Panel A** displays the contingency scores, ordered left to right, from the most to least significance. **Fig 9 Panel B** displays the SOM$_{DTP}$ projections for these significant defective genes. Projections, colored according the legend, represent instances where significant Student's t-statistics co-project with compounds having these MOAs. For example, *ABL1* projections (blue) appear mainly in the lower right region. Color coding is unique for all defective genes analyzed herein; intended to provide a visual separation for each defective gene. **Fig 9 Panel C** lists the SOM$_{DTP}$ node counts, ordered from top to bottom and left to right. **Fig 9 Panel D** displays a histogram for these counts.

The most frequently appearing defective genes are *MMP1*; associated mainly with MOAs PK, AM and Ho, and *NF1*; associated with MOAs PK, BCR-ABL, PARP and Ho. MOA:PK occurs most frequently with *MMP1*, *NF1* and *PIK3R1* as the most frequent defective genes. The second highest count is for MOA:Ang, which is associated with defective genes *RPTOR*, *SOX9* and *MTOR*. The next most frequent counts are associated with MOA:HO(*MMP1*, *NF1*, *PIK3R1*, *NCOR2* and *ABL1*), MOA:PARP(*NF1*, *NCOR2*, *PIK3R1* and *ACVR2A*), MOA:AM (*MMP1* and *ZHX2*) and MOA:BCR-ABL(*NF1* and *PIK3R1*). Inspection of **Fig 9 Panel D** summarizes these results. For example, MOA:PK has *MMP1* (teal) and *NF1* (light red) as representing the majority of co-projections, while MOA:BCR-ABL is dominated by *NF1* (light red).

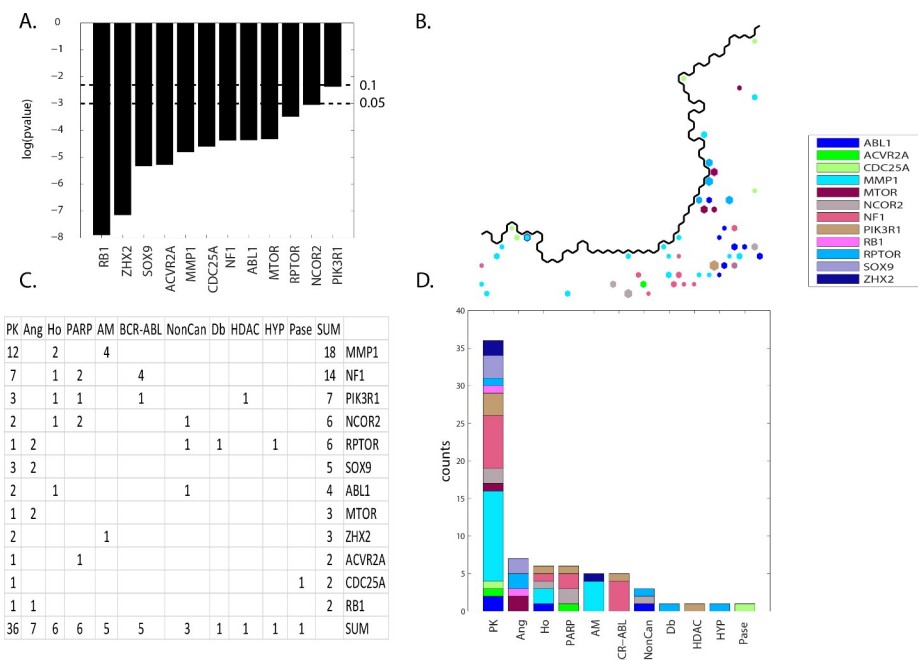

**Fig 9. Panel A displays the contingency scores, ordered left to right, from the most to least significance.** The horizontal dashed lines represent significance thresholds of $p < = 0.05$ (lower line) and $p < = 0.1$ (upper line). **Panel B** displays the $SOM_{DTP}$ co-projections of significant defective genes and MOAs for FDA compound. Only co-occurrences for $SOM_{DTP}$ projections of FDA compounds are displayed. The $SOM_{DTP}$ region displayed in **Panel B** represents the boundary for meta-clades 1 through 6 (see the white border in **Fig 8 Panel A**). **Panel C** lists the counts for co-occurrence (see **S6 master_appendix sheet gp_A**). **Panel D** displays the tabular results in **Panel C** as a histogram. Node colors for defective genes correspond to the legend inserted into the upper left panel. The counts displayed in **Panel C** represent the top 10th percentile of $SOM_{DTP}$ co-projections for FDA compounds. A consistent coloring scheme is used for this and all subsequent figures, such that all defective genes presented in the **RESULTS** are assigned a unique color. **S13 master_appendix_sheet gp_A_FDA** list the counts for each FDA and MOA entry for these significant genes.

The common feature of the defective genes associated with SOM meta-clade Group **A** is their potential to influence the Ras/Raf/MEK/ERK and the PI3K/AKT pathways. The Ras/Raf/MEK/ERK mitogen activated protein kinase (MAPK) cascade is constitutively active and is the dominant pathway driving the production *MMP1* [31], the defective gene with the highest $SOM_{DTP}$ node count. *MMP1* also modulates cytoskeleton organization, cell motility and additional metastasis signature genes [32] which in turn are mediated by the ERK pathway [33]. In general, the expression of the family of matrix metalloproteinases (MMP) is broadly affected by intracellular signaling via the MAPK family. Targeting the RAF-MEK-ERK mitogen-activated protein kinase cascade is being actively pursued for the treatment of cancer [34].

A direct role of *MMP1* on chemosensitivity has not been reported. However, Zhou et al. [35] identify *MMP1* as a potential gene conferring resistance of EGFR drugs targeting in non-small cell lung cancer. Rapamycin significantly enhanced the expression of interstitial collagenase (*MMP1*) at the protein and mRNA levels [36]. An assessment of upregulated expression levels in serous ovarian cancer cell lines by Zhang et al. [37] find matrix metalloproteinase 1 (*MMP1*) to be among the most upregulated mRNAs in the chemoresistant cell lines. Given that *MMP1* is the most frequent defective gene associated with MOA:PK (cf. **Fig 9**), combined with its role in chemosensitivity, suggests that defective *MMP1* may play a role in the weak $\underline{GI50}_{NCI60}$ responses to *PIK3* and *EGFR* targeting agents screened in the NCI60.

*NF1* has the 2nd highest node count in group **A** and has links to the MAPK cascade. For example, loss of *NF1* gene expression leads to increased RAS activation and hyperactivation of the downstream RAS effectors, including the RAF/MEK/ERK and the PI3K/AKT pathways [38]. Abnormal activation of RAS by defective *NF1* is a central driver event in some soft-tissue sarcomas (MPNST). Receptor tyrosine kinases (RTKs), including PDGFRA and EGFR, can activate RAS signaling and downstream factors such as MEK and mTOR. Ki et al. [39] find the addition of mTOR inhibitors to cell lines harboring defective *NF1* enhance the activity of DNA targeting agents. Defective genes that impact PI3K-Akt-mTOR signaling could weaken the tumor cell and enhance susceptibility to chemotherapeutic drugs.

A noteworthy entry in **S6 master_appendix sheet gp_A_FDA** is for Olaparib, MOA:PARP and defective gene *NF1*. Combination treatment with olaparib and various inhibitors *of PD-L1*, *VEGFR*, *PI3K*, and *AKT* may effectively inhibit the growth of rapidly proliferating triple negative breast cancer cell lines [40]. A review of candidate synthetic lethality partners to *PARP* inhibitors in the treatment of ovarian clear cell cancer by Kawahara et al. [41] finds *PARP* and *NF1* to be synthetic lethality pairs [42]. Synthetic lethality (SL) describes the genetic interaction by which the combination of two separately non-lethal mutations results in lethality [43]. Generally, the ablation of two genes located in parallel pathways (leading to cell survival or a common essential product) is one of the important patterns causing synthetic lethality. Synthetic lethality appears to be achieved with combined *EGFR* and *PARP* inhibition [44]. SL has recently emerged as a promising new approach to cancer therapy [45].

MOA:Ang ranks 2nd among the MOA's listed for group **A** and is associated with defective *RPTOR*, *SOX9* and *MTOR*. Oncogenic activation of the phosphatidylinositol-3-kinase (*PI3K*), and mammalian target of rapamycin (*MTOR*) facilitates tumor formation, disease progression, therapeutic resistance, and the sensitivity of prostate cancer cell lines to PI3K-AKT-mTOR-targeted therapies [46]. *SOX9* is reported to promote of tumor growth, proliferation, migration and invasion and the metastasis and regulation of Wnt/β-catenin signalling [47]. Inhibition of *SOX9* expression in led to a significant reduction in primary tumor growth, angiogenesis, and metastasis [48]. The full extent of the PI3K-AKT-mTOR signaling network during tumor/angiogenesis, invasive progression and disease recurrence remains to be determined. The existing results link chemosensitivity of MOA:Ang agents to a selective set of defective genes.

## Results: Group B(meta-clades 7 through 9)

Eleven MOA classes (PK, Ho, Db, NonCan, Ds, Apo, AM, T2, A7, HDAC, and PARP) are associated with eight defective genes (*ZHX2*, *MECOM*, *MMP1*, *EP300*, *MTOR*, *BMP7*, *CYP11B1* and *CYP11B2*) for SOM meta-clades 7 through 9. *ZHX2* has the most and *EP300* the least significant contingency scores (**Fig 10 Panel A**). *ZHX2* projects to the central region of group **A**, while *CYP11B2*, *MECOM* and *MMP1* project to the perimeter regions (**Fig 10 Panel B**). **Fig 10 Panels C** and **D** indicate that defective genes *ZHX2*, *MECOM*, *MMP1* and *PTK2* and MOAs PK, Ho and Db occupy the most $SOM_{DTP}$ nodes. These defective genes are associated with the GO ontology pathway Regulation_of_Response_to_Stress, with a potential to influence cellular functions such as differentiation and translation. *ZHX2* is a member of the zinc fingers and homeoboxes gene family that acts as a transcriptional repressor. Ontology (GO) annotations related to *ZHX2* also include DNA-binding_transcription_factor_activity. *MECOM (*MDS and EVI1 complex locus protein), with the 2nd highest $SOM_{DTP}$ counts, is found to be commonly enriched in cancer cell lines. Makondi et al. [49] suggest that targeting the MAPK signal transduction pathway through the targeting of the *MECOM* might increase tumor responsiveness to irinotecan treatment. Saito et al. [50] notes that EVI1 alters metabolic programming associated with leukemogenesis and increases sensitivity to L-asparaginase. The

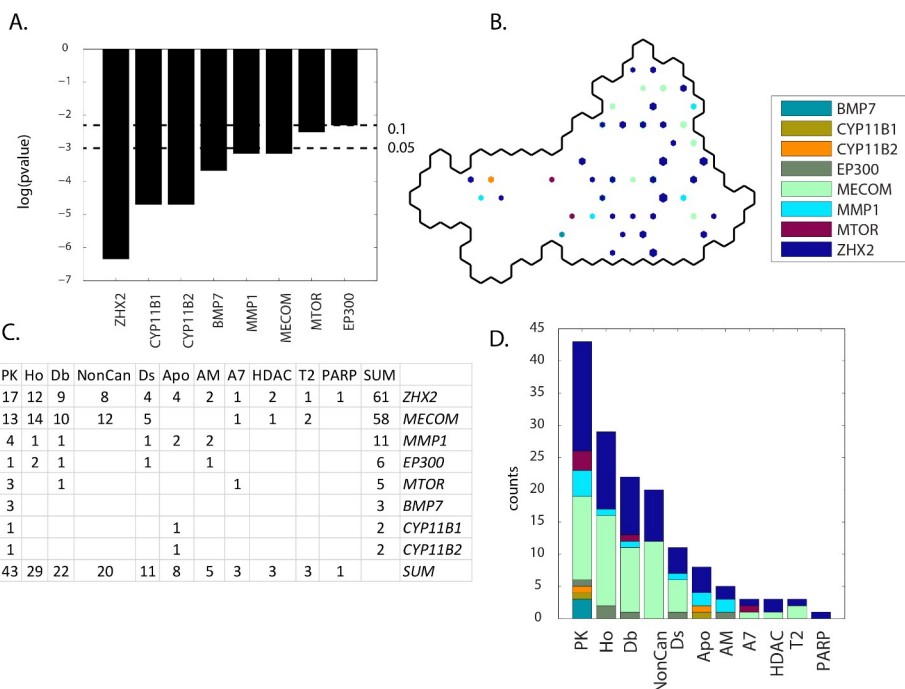

**Fig 10. Results for group B(meta-clades 7 through 9).** The SOM$_{DTP}$ region displayed in **Panel B** represents the boundary for meta-clades 7 through 9 (see the white border in **Fig 8 Panel A**). **S7 master_appendix sheet gp_B** lists the table in **Panel C**. See the legend of **Fig 9** for details. **S14 master_appendix sheet gp_B_FDA** lists the FDA compounds associated with these defective genes.

3$^{rd}$ most frequent gene, *MMP1*, is in the family of matrix metalloproteinases that are involved in the breakdown of extracellular matrix and contribute to metastasis, as noted above. **S7 master_appendix sheet gp_B (meta-clades 7 through 9)** lists the defective genes with a significant contingency score (p<0.1) for each meta-clade in Group **B**. Row entries in **S14 master_appendix sheet gp_B_FDA** list the counts for each FDA and MOA entry for significant defective genes.

## Results: Group C (meta-clades 10 through 15)

Contingency scores order the defective genes as: *NOTCH1*, *RBPJ*, *IGF1R*, *PIK3CG*, *CDKN2A*, *ATM*, *NRAS*, *MSH2*, *CDKN2B*, *CDC25A*, *NCOR2*, *RPTOR*, *STAT2*, *EIF5A2*, *MYC*, *SPEN* and *MTOR* (**Fig 11 Panel A**). *IGF1R* projects mainly at the perimeter of SOM$_{DTP}$ for group **C**, while the remaining defective genes project to more central regions (**Fig 11 Panel B**). Seventeen MOA classes, ordered from most to least node counts, are Ds, PK, HDAC, Apo, Ho, AM, BCR-ABL, A7, NFkB, BRD, Mito, NonCan, PARP, KLF4, PSM, T1 and SMO (**Fig 11 Panel C and D**).

The ten most frequent defective genes, *IGF1R*, *CDC25A*, *NOTCH1*, *NCOR2*, *RPTOR*, *CDKN2A*, *MSH2*, *NRAS* are associated with MOAs Ds, PK, HDAC, Apo, Ho, AM and BCR-ABL. The salient feature of these defective genes is their role in arresting the cell cycle. Cellular processes involving phosphorylation function to interrupt the cell-cycle, particularly from members of the family of tyrosine kinases. For example insulin-like growth factor 1 receptor (*IGF1R*) belongs to the large family of tyrosine kinase receptors and is activated by a hormone called insulin-like growth factor 1 (IGF-1) and by a related hormone called IGF-2 [51]. SOM$_{DTP}$ nodes in meta-clades 10 through 15 that are associated with defective *IGF1R*

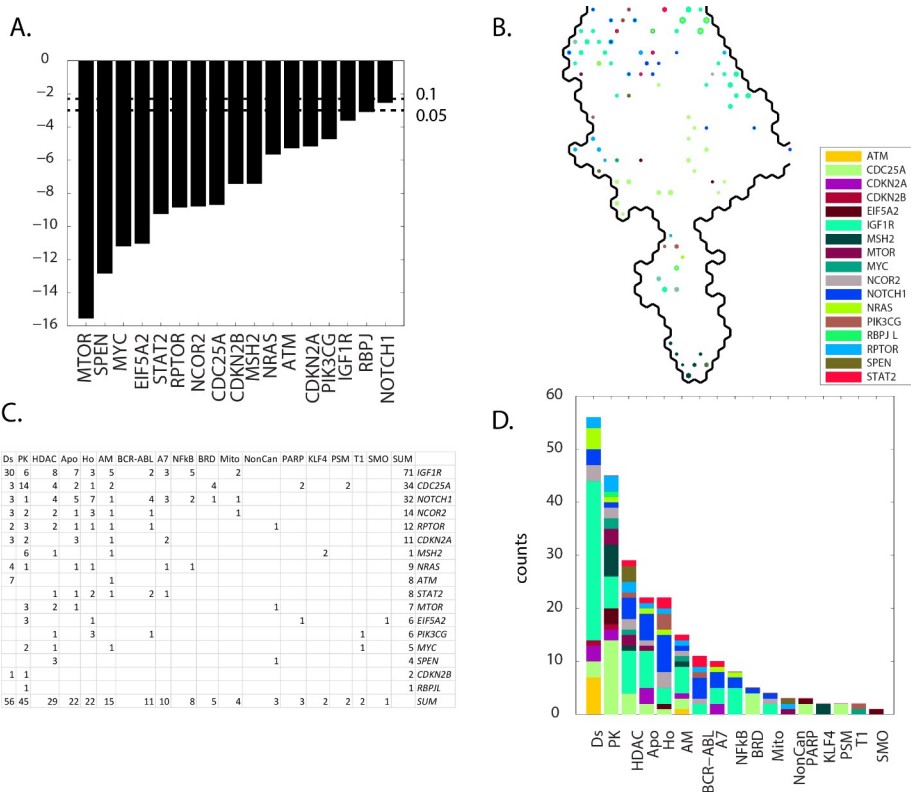

**Fig 11. Results for group C(meta-clades 10 through 15).** The SOM$_{DTP}$ region displayed in **Panel B** represents the boundary for meta-clades 10 through 15 (see the white border in **Fig 8 Panel A**). **S8 master_appendix sheet gp_C** lists the table in **Panel C**. See the legend of **Fig 9** for details. **S15 master_appendix sheet gp_C_FDA** lists the FDA compounds associated with these defective genes.

exist for chemosensitivity mainly to leukemia cell lines. *IGF1R* is often overexpressed by tumors and mediates proliferation and apoptosis protection [52,53]. As noted earlier [27], evaluation of drug sensitivity for compounds targeting leukemia cell lines has prompted the emergence of *IGF1R* as a potential therapeutic target for the treatment of leukemia. Weisberg, et al. [54] report that *IGF1R* protein expression/activity was substantially increased in mutant RAS-expressing cell lines, and suppression of RAS led to decreases in *IGF1R*. Synergy between *MEK* and *IGF1R* inhibitors correlated with induction of apoptosis, inhibition of cell cycle progression, and decreased phospho-S6 and phospho-4E-BP1. They suggested that given the complexity of RAS signaling, it is likely that combinations of targeted agents will be more effective than single agents, inclusive of *IGF1R* inhibitors.

*CDC25A*, with the 2nd highest node counts, affects cell proliferation and its expression is thought to be controlled through the PI3K-AKT-MTOR signaling pathway [55]. Sadeghi et al. [56] suggest that *CDC25A* controls the cell proliferation and tumorigenesis by a change in expression of proteins involved in cyclin D1 regulation and G1/S transition. The finding that defective *CDC25A* is associated with MOA:PK is consistent with the appearance of pazopanib and axitinib in the FDA compounds listed in **S15 master_appendix sheet gp_C_FDA**.

The evolutionarily conserved NOTCH family of receptors regulates a myriad of fundamental cellular processes including development, tissue patterning, cell-fate determination, proliferation, differentiation and cell death [57]. The crosstalk among *Notch1* (3rd highest node counts) and other prominent molecules/signaling pathways includes DNA damage repair

(DDR) [58]. DDR is a complex protein kinase based signaling pathway which is conducted by the members of the phosphoinositide 3-kinase-like kinase (PIKK) family, such as ataxia telangiectasia mutated (*ATM*). *NOTCH1* is a major oncogenic driver in T cell acute lymphoblastic leukemia [59]. *NOTCH1* siRNA can effectively inhibit the expression of *NOTCH1* gene, inhibit the proliferation of lung cancer A549 cell lines and increase the sensitivity to chemotherapeutic drugs [60]. Of specific interest is the intersection of defective *NOTCH1* and the projection for imatinib (**S15 master_appendix sheet gp_C_FDA**). Aljedai et al. [61] explored the role of *NOTCH1* signaling in chronic myeloid leukemia cell lines to find cross-talk between *NOTCH1* and *BCR-ABL*. Their results revealed that imatinib induced *BCR-ABL* inhibition results in upregulation of *NOTCH1* activity. In contrast, inhibition of *NOTCH1* leads to hyperactivation of *BCR-ABL*. They proposed that the antagonistic relationship between *NOTCH1* and *BCR-ABL* in CML suggests a combined inhibition of *NOTCH1* and *BCR-ABL* may provide superior clinical response over tyrosine-kinase inhibitor monotherapy.

   *CDKN2A*, *MSH2* and *ATM* (with the next most frequent node counts) have roles in cell cycle arrest. *CDKN2A* is capable of inducing cell cycle arrest in the G1 and G2 phases. Gene Ontology (GO) annotations related to *CDKN2A* include transcription factor binding. *MSH2* and *ATM* are components of the post-replicative DNA mismatch repair system (MMR), whereby activation of checkpoint arrest and homologous DNA repair are necessary for maintenance of genomic integrity during DNA replication [62]. Germ-line mutations of the ataxia telangiectasia mutated (*ATM*) gene result in the well-characterized ataxia telangiectasia syndrome, which manifests with an increased cancer predisposition. Somatic *ATM* mutations or deletions are commonly found in lymphoid malignancies. Such mutations may be exploited by existing or emerging targeted therapies that produce synthetic lethal states. Cancers with mutations in genes encoding proteins involved in DNA repair may be more sensitive to treatments that induce synthetic lethality by inducing DNA damage or inhibiting complementary DNA repair mechanisms.

## Results: Group D(meta-clades 16 through 18)

Contingency scores order the defective genes as: *MAP2K3*, *PTK2*, *BRAF*, *CYP11B1*, *CYP11B2*, *MMP9*, *MYC*, *FLT1* and *RBPJL* (**Fig 12** **Panel A**). *MYC*, *BRAF* and *FLT1* project to mainly non-overlapping $SOM_{DTP}$ regions (**Fig 12** **Panel B**). Eight MOA classes, ordered from most to least node counts, are Tu, HSP90, NonCan, PSM, DB, T1, PK, T2 and Pase (**Fig 12** **Panels C** **and D**). MOA:Tu dominates these results, while MOA:HSP90, MOA:NonCan and MOA:PSM appear with the next highest node counts. The most frequent defective genes include *MYC*, *RBLJL*, *FLT1* and *MMP9*. **S16 master_appendix sheet gp_D_FDA** lists the FDA compounds associated with these defective genes.

   Most of the defective genes in group **D** are involved with the mitotic component of tumor cell proliferation. For example, *MYC* encodes a nuclear phosphoprotein that has been implicated in the regulation of cell proliferation and the development of human tumors [63] and is regarded as a major determinant of mitotic cell fate [64]. Inhibition of microtubule polymerization has been reported to block mitosis and induce cell death [65]. Conacci-Sorell et al. [66] report the expression of *MYC* results in the induction of the actin-bundling protein fascin, formation of filopodia, and plays a role in cell survival, autophagy, and motility. *MYC* also recruits acetyltransferases that modify cytoplasmic proteins, including α-tubulin. Marzo-Mas et al. [67] find the antiproliferative activity of colchicine to inhibit tubulin polymerization to be modulated by the downregulation of *c-MYC* expression. Alexandrova et al. [68] report that the N-terminal domain of *c-MYC* associates with alpha-tubulin and microtubules. Marzo-Mas et al. [67] also found that tubulin binding compounds were able to downregulate the

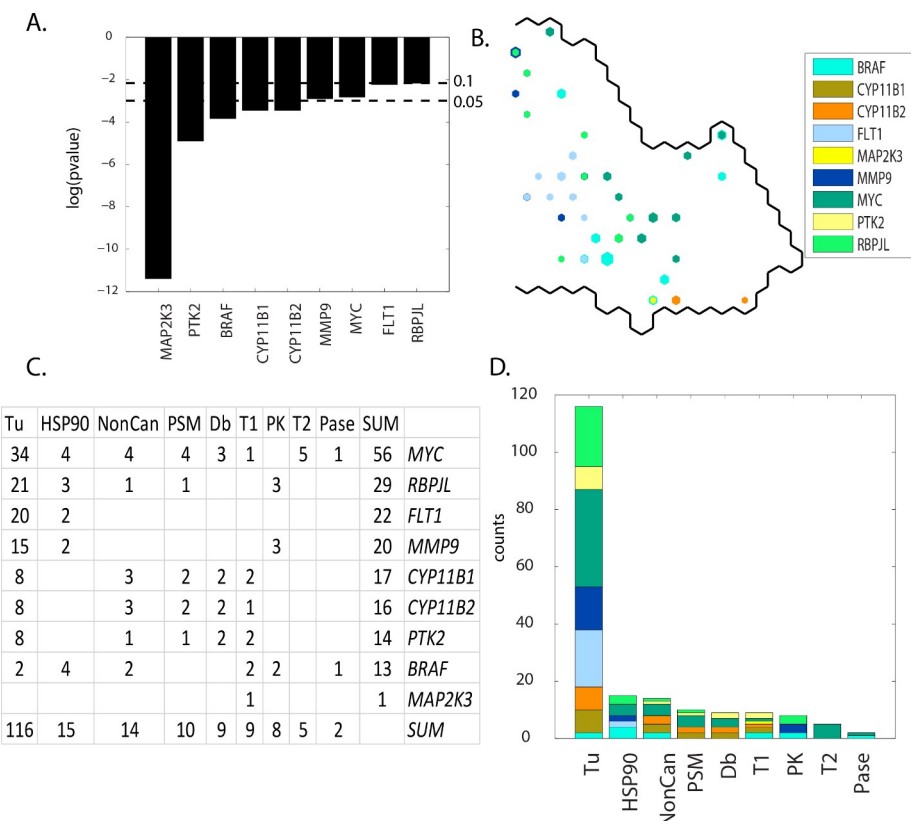

**Fig 12. Results for group D(meta-clades 16 through 18).** The SOM_{DTP} region displayed in **Panel B** represents the boundary for meta-clades 16 through 18 (see the white border in **Fig 8 Panel A**). **S9 master_appendix sheet gp_D** lists the table in **Panel C**. See legend of **Fig 9** for additional details. **S16 master_appendix sheet gp_D_FDA** lists the FDA compounds associated with these defective genes.

expression of the VEGF, hTERT and *c-MYC* genes. Others [69] have proposed targeting oncogenic *MYC* as a strategy for cancer treatment, proposing the destruction of a microtubule-bound *MYC* reservoir during mitosis contributes to vincristine´s anti-cancer activity [70]. Collectively these results support a role of defective *MYC* in chemosensitivity to tubulin targeting agents.

The 2nd most frequent defective gene is *RBPJL*. *RBPJL* binds to DNA sequences almost identical to that bound by the Notch receptor signaling pathway transcription factor recombining binding protein J (*RBP-J*). A related family member RITA (RBPJ Interacting And Tubulin Associated 1) also acts as a negative regulator of the Notch signaling pathway that induces apoptosis and cell cycle arrest in human hepatocellular carcinoma [71]. Structural and biophysical studies demonstrate that RITA binds *RBP-J* and biochemical and cellular assays suggest that RITA interacts with additional regions on *RBP-J* [72]. Emerging evidence reveals Notch as a microtubule dynamics regulator and that activation of Notch signaling results in increased microtubule stability [73]. The *RBPJL/RITA* association raises the possibility that RITA-mediated regulation of Notch signaling may be influenced by *RBPJL* and potentially play a role in the chemosensitivity of Tu agents.

The 3rd most frequent defective gene is *FLT1* (Fms-related tyrosine kinase (FLT) or VEGF receptor 1). The role of *FLT1* in the chemosensitivity of tubulin agents would appear to be unexpected. However, the blockade of VEGFR-1 and VEGFR-2 enhances paclitaxel sensitivity in gastric cancer cell lines [74]. Microtubule-targeted drugs inhibit VEGF Receptor-2

expression by both transcriptional and post-transcriptional mechanisms [75]. Novel anti-mitotics, which target the mitotic spindle through interactions with non-microtubule mitotic mediators like mitotic kinases and kinesins, have been identified and are now in clinical testing [76]. Included in clinical testing are compounds that have low nanomolar potency against *ABL*, *FLT1* and *PDGFR* [77]. Tumor endothelial cell lines demonstrate a strong activation of VEGF and Notch signaling [78]. VEGF-B is a growth factor that binds *FLT1* and is considered the odd member of the VEGF family, with mainly angiogenic and lymphangiogenic activities. VEGF-B has protective effects on neuropathy [79]. *FLT1* has been proposed as a prognostic indicator in endometrial carcinoma [80].

The 4th most frequent defective gene, *MMP9* (matrix metalloproteinases 9) and its associated vascular endothelial growth factor (VEGF) are critical for tumor vascularization and invasion. A recent study of the expression of *MMP-9* and VEGF(*FLT1*) in breast cancer patients found their correlation significant enough to propose these genes as prognostic indicators [81]. Inspection of these SOM meta-clades finds MOA:PK agents to be located mainly in the upper portion of SOM meta-clades 16 through 18, where defective genes *MMP9* and *RBPJL* also appear. Crizitonib is co-projected to these SOM nodes. Cizitonib is a small molecule TKI that inhibits the activity of the *ALK* fusion proteins, *MET*, *ROS1*, and *MST1R* (RON) [82,83]. Noteworthy is the impressive clinical responses to crizotinib and other small-molecule drugs inhibiting the *ALK* tyrosine kinase [84]. Defective *MMP9* or *RBPLJ* may contribute to enhanced crizitonib chemosensitivity.

MOA:HSP90 is the 2nd most frequent MOA class in SOM meta-clades 16 through 18. Several studies have suggested a possible connection between HSP90 and the microtubule cytoskeleton. Weis et al. [85] find that HSP90 protects tubulin against thermal denaturation. Antitumor selectivity of a novel Tubulin and HSP90 dual-targeting inhibitor has been identified in non-small cell lung cancer model [86]. The presence of geldanamycin within the list of agents in this SOM region is consistent with this observation. Liu et al. ([87]) find evidence that misregulated HSP90 can affect drug sensitivity, an effect proposed to be due the altered regulation of HSP90 client proteins, inclusive of tubulin.

## Group E(meta-clades 19 through 20)

Contingency scores order the defective genes as: *BRAF*, *EGFR*, *JAK3*, *RPTOR*, *PIK3CG* and *SPTBN1* (**Fig 13** **Panel A**). *BRAF* dominates the central region of SOM$_{DTP}$ for group **E** (**Fig 13** **Panel B**). Nine MOA classes, ordered from most to least node counts, are Pk, BCR-ABL, HDAC, NonCan, PSM, A7, Ds, HSP90 and Ho (**Fig 13** **Panels C** and **D**). The most frequent defective gene is BRAF and is associated with the most frequent MOA:PK, followed by MOA: BCR-ABL, MOA_HDAC. *PIK3CG*, *RPTOR* and *JAK3* are the 2nd, 3rd and 4th ranking defective genes. This SOM$_{DTP}$ region corresponds to the projection of known FDA approved *BRAF* targeting agents; dabrafenib, hypomethicin, selmutinub and vemurafenib (**S17 master_appendix sheet gp E_FDA**). These results are consistent with the findings of Ikediobi et al. [4].

The association of defective *BRAF* with compounds that target this condition are well documented [4,88]. Mutant *BRAF* (v-Raf murine sarcoma viral oncogene homolog B1) inhibitors such as vemurafenib and dabrafenib have achieved unprecedented clinical responses in the treatment of melanomas [89,90]. The association of defective *BRAF* to MOA:HDAC is consistent with literature reports. Recent studies have shown that histone deacetylase (HDAC) and mutant *BRAF* (v-Raf murine sarcoma viral oncogene homolog B1) inhibitors synergistically kill melanoma cell lines with activating mutations in *BRAF* by induction of necrosis [91].

A role for defective *PIK3CG* is indicated in SOM meta clades 19 through 20 for MOA:PK and MOA:BCR-ABL. The publications from Shi et al. [92], Van Allen et al. [93] and Rizos

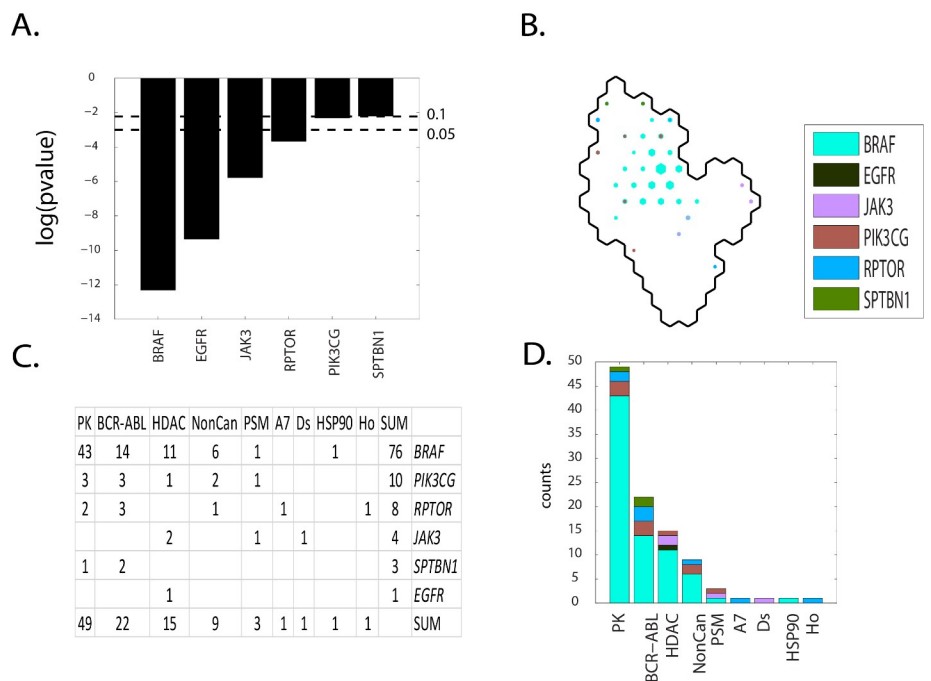

**Fig 13. Results for group E(meta-clades 19 through 20).** The SOM$_{DTP}$ region displayed in **Panel B** represents the boundary for meta-clades 19 through 20 (see the white border in **Fig 8 Panel A**). **S10 master_appendix sheet gp_E** lists the table in **Panel C**. See legend to **Fig 9** for additional details. **S17 master_appendix sheet gp_E_FDA** lists the FDA compounds associated with these defective genes.

et al. [94] addressed the roles of *PI3K* pathway gene's mutations. Resistance to *BRAF* inhibitors can be associated with upregulation of the PI3K/AKT pathway, resulting from AKT1/3 mutations and mutations in positive (*PIK3CA*, *PIK3CG*) and negative (*PIK3R2*, *PTEN* and *PHLPP1*) regulatory genes [95]. The results in **Fig 13 Panels C** and **D** indicate a role for HDAC in *BRAF* chemosensitivity. Gallagher et al. [96] find that HDAC inhibitors affect *BRAF*-inhibitor sensitivity by altering *PI3K* activity.

A role for defective *RPTOR* is indicated for MOA:BCR-ABL. Drugs simultaneously targeting two or more pathways essential for cancer growth could slow or prevent the development of resistant clones. Puausova et al. [97] identify dual inhibitors of proliferative pathways in human melanoma cell lines bearing the V600E activating mutation of *BRAF* kinase. They found these inhibitors to simultaneously disrupt the *BRAF* V600E-driven extracellular signal-regulated kinase (ERK) mitogen-activated protein kinase (MAPK) activity and the mechanistic target of rapamycin complex 1 (mTORC1) signaling in melanoma cell lines, yielding dynamic changes in mTOR(*RPTOR*) signaling.

The non-receptor tyrosine Janus kinases (JAK) are involved in various processes such as cell growth, development, or differentiation. The result presented here finds an enhanced chemosensitivity to HDAC inhibitors for tumor cell lines with defective *JAK3*. DUAL kinase and HDAC inhibitors have been developed based on the reasoning that specifically blocking more than one oncogenic pathway simultaneously with a combination of different drugs may be a more effective cancer treatment [98]. Dual inhibitors of Janus kinases and HDAC have been developed [99,100], As an example, Dymock's group has designed and synthesized a novel series of dual JAK and HDAC inhibitors based on the core features of ruxolitinib [101]. Upregulation of *JAK3* has been observed in response to increases of oxygen-containing species following HDAC inhibition [102]. Although the design of dual JAK/HDAC inhibitors was based

on simultaneously targeting different oncogenic pathways, a role for defective *JAK* in chemosensitivity may be important.

## Results: Group F(meta-clades 21 through 24)

Contingency scores order the defective genes as: *AXL, STAT2, CDKN2A, RPTOR, KRAS and MYC*(**Fig 14** **Panel A**). *CDKN2A* is the dominant defective gene in meta-clade 21 while *RPTOR* and *STAT2* are located primarily in meta-clade 22 (**Fig 14** **Panel B**). Eighteen MOA classes exist, with MOAs appearing with the highest counts all targeting DNA (Ds, T2, A7, Db, Df and A2)(**Fig 14** **Panels C** and **D**). *CDKN2A* as the most frequent defective gene, followed by *AXL, MYC, STAT2 and KRAS*. Meta-clades 21 through 24 represent, by far, the largest number of FDA approved agents. These defective genes affect proliferation largely resulting from their role in targeting DNA, DNA damage repair and phosphorylation. These genes do not overlap with a prior analysis of DNA repair genes in the NCI60 and their predictive value for anticancer drug activity [103].

Su et al. [104] report that *CDKN2A* loss is significantly associated with the sensitivity of CDK4/6 inhibitors (also projected to SOM meta-clade 14). Evidence supports the role of *CDKN2A* in cell cycle independent functions such as DNA damage repair [105]. *CDKN2A* also provides instructions for making several proteins, including p16(INK4A) and p14(ARF), which function as tumor suppressors that keep cell lines from growing and dividing too rapidly or in an uncontrolled way. Overexpression of *CDKN2A* inhibits cell proliferation and invasion, to cause cell cycle arrest in the G1 phase. *CDKN2A* mediates the AKT–mTOR (*RPTOR*) signaling pathway by suppressing lactate dehydrogenase (LDHA) [106]. Taken together, these results suggest therapeutic agents that target *CDKN2A* and *RPTOR* **i**n cancers that share these defective genes. Consistent with chemosensitivity for FDA compounds in these meta-clades, recent observations report that long term survivorship after high dose DNA

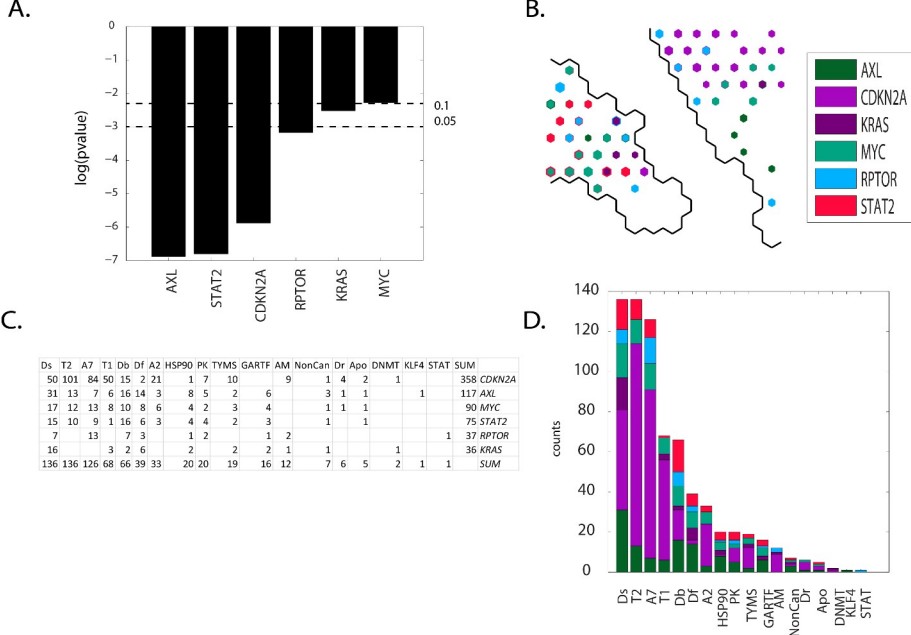

**Fig 14. Results for group F(meta-clades 21 through 24).** The SOM_DTP region displayed in **Panel B** represents the boundary for meta-clades 21 through 24 (see the white border in **Fig 8** **Panel A**). **S11 master_appendix sheet gp_F** lists the table in **Panel C.** See legend to **Fig 9** for additional details. **S18 master_appendix sheet gp_F_FDA** lists the FDA compounds associated with these defective genes.

damaging chemotherapy with melphalan is compatible with an increased chemosensitivity due to impairment of the DNA repair pathway [107]. Loss of *CDK2* presents a different challenge to cell lines, aside from the more conventional role to regulate cyclins, which in turn might lead to altered DNA damage response and checkpoint activation, mutations in DNA repair genes drive cancer development [108,109]. Ras proteins play a crucial role as a central component of the cellular networks controlling a variety of signaling pathways that regulate growth, proliferation, survival, differentiation, adhesion, cytoskeletal rearrangements and motility of a cell [110]. *KRAS* (Kirsten-rat sarcoma viral oncogene homolog) is a prominent oncogene that has been proven to drive tumorigenesis, modulate numerous genetic regulatory mechanisms including the induction of DNA damage repair pathways [111,112]. Mutant RAS-driven tumorigenesis arises independently of wild-type RAS isoforms, but recent evidence indicates wild-type isoforms are involved. Grabocka and colleagues [113] report how the loss of wild-type RAS alters oncogenic signaling and dampens the DNA-damage response, thereby affecting tumor progression and chemosensitivity. Since the MOA agents listed for SOM meta-clades 21 through 24 have roles in DNA damage, defective *CDKN2A*, *RPTOR* and *KRAS* may contribute to chemosensitivity of tumor cell lines to these agents. While targeting defective *KRAS* remains elusive [114], small molecule inhibitors are in the pipeline [115]. Exploration of NCI60 screened compounds that project to meta-clades 21 through 24 may provide a starting point for lead discovery.

Pyrazoloacridine, palbociclib, methotrexate, fluorouracil, 8-Chloro-adenosine, pralatrexate, pemetrexed, pelitrexol, by-product_of_CUDC-305, 6-Mercaptopurine and oxaliplatin appear most frequently in the SOM region for group **F (S18 master_appendix sheet gp_F_FDA)**. A study of gastric cancer patients detected a high frequency of mutations in *MLL4*, *ERBB3*, *FBXW7*, *MLL3*, *mtor(RPTOR)*, *NOTCH1*, *PIK3CA*, *KRAS*, *ERBB4* and *EGFR* [116]. *KRAS* mutations have been reported as predictors of the response of lung adenocarcinoma patients receiving platinum-based chemotherapy [117,118]. *NOTCH1* mutations target *KRAS* mutant CD8+ cells to contribute to their leukemogenic transformation [119,120]. Notable in the list of FDA approved agents associated with SOM meta-clades 21 through 24 is oxaliplatin. Oxaliplatin-based chemotherapy is more beneficial in *KRAS* mutant than in *KRAS* wild-type metastatic colorectal cancer [121]. SOM meta-clade 21 is the location of cytarabine (ara-C) and is consistent with the conclusion of Ahmad et al [24] that adult AML patients carrying defective *KRAS* benefit from higher ara-C doses more than wt *KRAS* patients. Enhanced chemosensitivity of tumor cell lines with defective *KRAS* may represent a link to these observations.

## Results: Group G(meta-clades 25 through 28)

Contingency scores order the defective genes as *PEG3*, *ABL1* and *PIK3CG* as the most significant and *MTOR*, *KRAS* and *NRAS* as the genes with the least significance (**Fig 15** **Panel A**). *PIK3CG* represents the largest count of SOM$_{DTP}$ projections, located mainly in the central region of meta-clades 25 through 28. *MTOR*, *KRAS* and *NRAS* are located mostly in the bottom of this region (**Fig 15** **Panel B**). Twelve MOA classes exist, with MOAs appearing with the highest counts as Ds, Apo, and PK (**Fig 15** **Panels C** and **D**). The most frequently occurring defective genes are *PIK3CG*, *NRAS*, *PTK2* and *ABL1* (**Fig 15** **Panels C** and **D**). The defective genes in meta-clade 25 through 28 represent an amalgamation of many of the previous meta-clade groups, where sets of defective genes were involved in cellular processes of phosphorylation and progression through the cell cycle for proliferation. Consequently, many of these defective genes have been previously discussed, with the exceptions of *NRAS* and *PTK2*.

*NRAS* (ranked 2nd by node counts) is one of the most common targets of oncogenic signaling mutations in hematologic malignancies. Even with the challenge of directly targeting

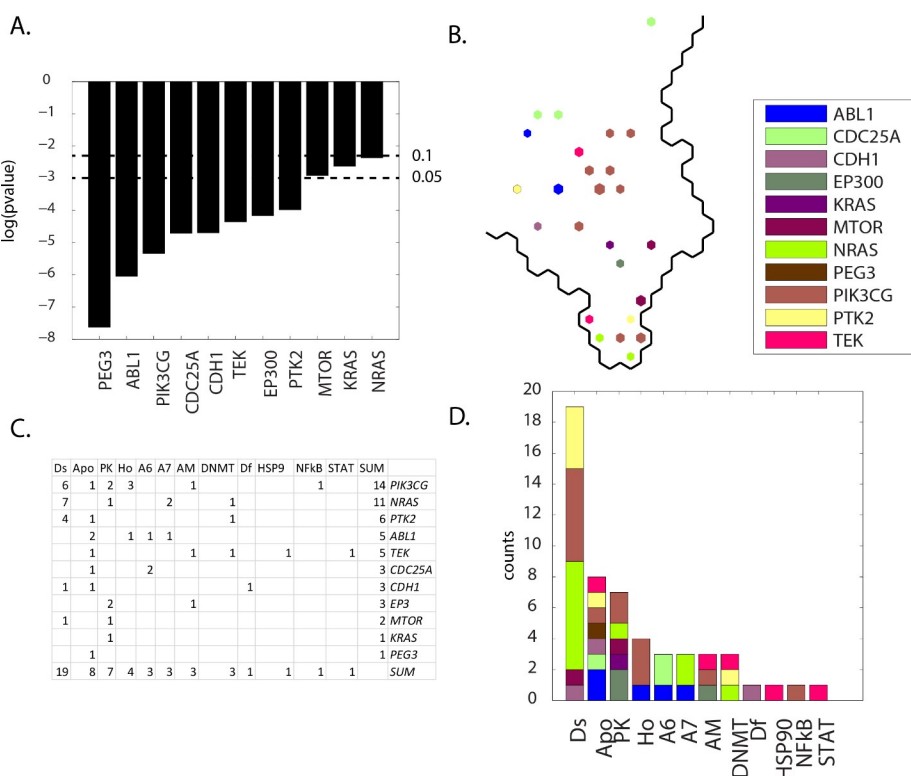

**Fig 15. Results for group G(meta-clades 25 through 28).** The SOM$_{DTP}$ region displayed in **Panel B** represents the boundary for meta-clades 25 through 28 (see the white border in **Fig 8 Panel A**). **S12 master_appendix sheet gp_G** lists the table in **Panel C.** See legend to **Fig 9** for additional details. **S19 master_appendix sheet gp_G_FDA** lists the FDA compounds associated with these defective genes.

mutant RAS oncoproteins, mitogen-activated protein kinase (MAPK) inhibition has been shown to reduce leukocytosis by targeting the downstream pathway of *NRAS* [122]. As noted earlier, combinations of targeted agents may not supersede conventional cytotoxic regimens, however combinations may enhance treatment efficacy. Identifying compounds that target defective *NRAS* and other compounds that target defective *PTK2*, *ABL1* or *PIK3CG*, in the case of meta-clades 25 through 28, may offer effective combination therapies. Without doubt, large numbers of molecular pathways are likely to be synergistically involved in cancer biology, the contribution of each pathway may be different and identifying which combinations to select will be experimentally exhausting. Bioinformatic approached as discussed herein may offer useful clues.

*PTK2* (ranked 3$^{rd}$ by node counts) is a non-receptor protein-tyrosine kinase with functions that include cell migration, reorganization of the actin cytoskeleton, cell cycle progression, cell proliferation and apoptosis through kinase-dependent and -independent mechanisms [123]. It is a member of the FAK (focal adhesion kinase) subfamily of protein tyrosine kinases and is listed as a transcriptional regulator. FAKs are reported to modulate chemosensitivity by altering chemokine production [31]. Enhanced chemosensitivity to gemcitabine has been reported with interference of FAKs [124]. Because of the involvement of *PTK2*(FAK) in many cancers, drugs that inhibit FAK are being sought and evaluated [125]. A screen to identify mechanisms of bleomycin resistance identified *Sky1*, *PTK2* and *Agp2* as determinants of chemosensitivity [126].

## Discussion

The development of rational strategies for targeted cancer therapy will require integrative analysis of data derived from diverse sources including, but not exclusive to, large-scale, publicly available, pre-clinical and clinical small-molecule screening and genomic data. A widely accepted challenge of linking screening and genomic data is how to gain molecular insight into the MOA(s) of active compounds. Not unexpectedly, the range of potentially important links is enormous; yielding massive challenges to the development of statistical/computational bioinformatic tools that assist integrative analyses. Advances have been made by focusing studies on fewer compounds (24 compounds in the CCLE [127] or approved FDA compounds [128]) or by studying small numbers of driver or mutated genes [129].

The results of the present study demonstrate the power of combining genomic data and small-molecule screens of FDA compounds in the NCI60 to provide mechanistic clues about compound activity. These results reveal coarse-grained associations between chemosensitivity of target-directed FDA agents towards tumor cell lines harboring specific genetic defects. SOM clustering finds seven regions of $GI50_{NCI60}$ responses, broadly assigned to FDA MOA classes that target, not exclusively, tubulin, BRAF mutations, RAF/MEK/ERK/mTOR and the PI3K/AKT pathways, DNA or protein synthesis pathways, the cell cycle and are associated with a relatively unique set of defective genes for each MOA class. Salient associations include the role of defective *MYC* for tubulin targeting agents, defective *CDKN2A*, *NRAS* and *KRAS* for DNA damaging/targeting agents and the role of defective *NOTCH1* for mutant BRAF targeting agents. Remarkably, nearly half of the defective genes reported herein also appear in Ikediobi et al. [4], albeit using very different methods.

The results described here may be applied to future pre-clinical studies. Notably there are exploitable instances of enhanced chemosensitivity of compound MOA's for a few defective genes. Specifically, there is support for synthetically lethal defective genes as contributing to chemosensitivity. Defective genes exist withing the NCI60 as doublets, triplets, quartets, etc., and a subset of these genes are associated with tumor cell lines that exhibit chemosensitivity. Exploiting chemosensitive $SOM_{DTP}$ nodes associated with tumor cell lines having more than one defective gene, that are also associated with numerous screened compounds, may identify additional synthetic lethal strategies. The notion of targeting parallel pathways can be extend beyond synthetically lethal genes. Combining agents with enhanced chemosensitivity against one defective gene, and its related cellular pathways, with other agents showing enhanced chemosensitivity towards other defective genes in alternative pathways, may enhance the efficacy of each agent. For example, each $SOM_{DTP}$ node with significant chemosensitivity for one defective gene includes many NSCs with similar $GI50_{NCI60}$ responses, inclusive of FDA compounds. Combinations of NSCs from $SOM_{DTP}$ nodes also exhibiting differential chemosensitivity for one or more defective gene in parallel pathways may be considered for experimental testing. The goal would be to identify combinations of NSCs that separately target parallel cellular pathways to determine whether their combination would enhance individual efficacies. The bioinformatic analysis described herein may provide clues for experimental pre-clinical testing of possible drug combinations.

Important caveats underly the interpretation of the results presented here. First, links of defective genes to chemosensitivity are not revealed in a clear-cut manner. Rarely is chemosensitivity associated only to tumor cell lines harboring defective genes. Chemosensitivity also exists within tumor cell lines lacking defective genes (cf. **Fig 3**). Consequently, while gene-drug associations may provide a genetic basis for drug selection [130], there is clear evidence herein that additional, not well understood, factors are in play. Second, combinations of defective genes appear to play a role in chemosensitivity. For example, 44 defective genes are listed

in the tables provided in each of the **RESULTS** subsections. Eighty-eight percent of these genes are listed only once (N = 24) or twice (N = 14) across the seven meta-clade groups. This result is an indication that relatively few defective genes contribute to enhanced chemosensitivity across meta-clade groups. In contrast, only two (*RPTOR* and *MTOR*) and three genes (*PIK3CG*, *MYC* and *CDC25A*) appear jointly in four or three of the seven meta-clade groups, respectively. Consequently, identifying a single defective gene as responsible for chemosensitivity may be rare; while combinations with genes commonly labeled as cancer genes may be more likely. Third, the 44 defective genes listed in the **RESULTS** subsections can be compared to current compendia of cancer gene mutations derived from human studies. The Cancer Genome Interpreter [131] has been developed to classify protein-coding somatic mutations and copy number variants into predicted passenger or known/predicted oncogenic mutations. Half (N = 22) of the defective genes listed here are identified by the Cancer Genome Interpreter's encyclopedia of patient-derived tumor xenografts (PDX) as driver mutations. A recent report using driver mutation patterns for prioritization of personalized cancer therapy [132] finds nearly 20% of their 39 tumor biomarkers to be included in this set of defective genes. Although the defective genes listed here were derived from novel applications of bioinformatic tools, these results find support within other databases. The absence of overlapping genes suggests potentially important roles for non-driver genes in chemosensitivity. Fourth, global analysis of modest to large scale genomic and screening data offers only one perspective. The genetic make-up of the NCI60 represents only a snapshot of data for a small number of tumor cell lines. The universal application of results derived from the NCI60 may be relevant only in the rare instance that another tumor cell matches the genetic makeup of any NCI60 cell. This does not, however, rule out analyses, parallel to that presented here, that jointly examine existing and new data. Fourth, the absence of defective *TP53* in these results has not gone undetected. Most NCI60 tumor cell lines harbor defective *TP53*. As a result, establishing a statistically significant Student's t-test for selective chemosensitivity fails mainly due to too few responses of tumor cell lines lacking defective *TP53*. Extending the data analysis to more tumor cell lines, lacking mutant *TP53*, may prove helpful. While addressing each of these caveats is massively challenging, resolution of each issue contributes to the understanding of preclinical screening results derived from a small set of human tumor cell lines.

In summary, the challenge of finding meaningful results within complex and noisy data has been proposed using contemporary data and state-ot-the-art statistical tools. This global analysis of multiple datasets, overlapping in their origins within the NCI60, has provided a unique perspective for associations of chemosensitivity, defective genes and MOAs.

## Supporting information

**S1 Fig.**
(TIF)

**S2 Fig.**
(TIF)

**S3 Fig.**
(TIF)

**S4 Fig.**
(TIF)

**S1 File. Manuscript_plos_revised_figs.**
(DOCX)

**S2 File. Manuscript_appendix_DTP.**
(DOCX)

## Acknowledgments

I would like to thank Drs. Ruili Huang and John Beutler for their extremely helpful comments provided during the preparation of this manuscript. I would also extend a special note of appreciation to the reviewers for their thoughtful suggestions.

## Author Contributions

**Conceptualization:** David G. Covell.

**Data curation:** David G. Covell.

**Formal analysis:** David G. Covell.

**Methodology:** David G. Covell.

**Software:** David G. Covell.

**Validation:** David G. Covell.

**Visualization:** David G. Covell.

**Writing – original draft:** David G. Covell.

**Writing – review & editing:** David G. Covell.

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
