## [Decision Letter · Decision Letter 0]

13 Jan 2021

PONE-D-20-35751

Bioinformatic Analysis Linking Genomic Defects to Chemosensitivity and Mechanism of Action

PLOS ONE

Dear Dr. Covell,

Thank you for submitting your manuscript to PLOS ONE. After careful consideration, we feel that it has merit but does not fully meet PLOS ONE’s publication criteria as it currently stands. Therefore, we invite you to submit a revised version of the manuscript that addresses the points raised during the review process.

We look forward to receiving your revised manuscript.

Kind regards,

Rama Krishna Kancha

Academic Editor

PLOS ONE

Journal Requirements:

Additional Editor Comments:

The reviewers are of the opinion that the methods section is not described enough for the study to be replicated. A detailed introduction and methods section is warranted. Further, all the changes in figures as well as in the results and discussion sections as suggested by the reviewers need to be included. The main findings of the study has to be highlighted clearly.

Comments from the Reviewer 1 (in case the author can't access the attached file):

OVERALL

This manuscript combines three data sources: drug response data on the NCI60 cell lines which is clustered using self-organizing maps, gene alteration data on the NCI60 cell lines, and annotations of the mechanism of action (MOA). The primary goal is to relate disrupted or defective genes to mechanisms of action using the clustered drug compounds as an intermediary. The methods do not appear to be particularly novel (various clustering techniques, t-tests, and Fisher's Exact Tests) and many of the identified genes are already well-known as drivers of cancer and as targets of specific therapies. Nevertheless, there is utility in terms of the scope of the analysis. Unfortunately, the methods are not described in adequate detail, and there appears to have been no adjustment to account for multiple testing. Thus, in its present form, it is not possible to fully tell if the results are actually correct. Moreover, the results section appears to contain a substantial amount of discussion. Finally, the results-discussion appear overly long, giving the impression of a "data dump" without adequate attention having been paid to highlight the most significant aspects of the findings.

MAJOR COMMENTS

[1] A fundamental problem with this manuscript is that it would be completely impossible for anyone to replicate the results. The author never explains what the DTP is. This reviewer happens to know that it is the Department of Therapeutic Programs at the NIH. Going to their web site and guessing that (1) I need to follow the “bulk downloads” link and (2) I need to follow the "NCI60 Growth Inhibition Data" link gets me to a page with another decision. Which release of this data was used? October 2020? June 2020? Something else? Downloading the October 2020 release, one learns a couple of things. First, it contains data on about 55K compounds, more than the 53K mentioned in the manuscript. Second, it contains data on 159 cell lines, not just the 59 that are part of the NCI60. Leaving the reader in the dark about exactly which data were used (even if it is publicly available) is unacceptable.

[2] In general, the methods are also described too vaguely and with too little detail to allow a reader to replicate the results.

[3] Was the CV computation used for filtering compounds based on the negative log-transformed GI50 data, or was it computed on some other scale?

[4] Please describe in more detail the heuristic that was used to decide on the size of the SOM grid, or provide a complete reference.

[5] The text twice mentions a manuscript by "Holbeck et al." The first time, it does not link to a reference in the bibliography. The second time, it links to reference #19, by Whyte and Holbeck. Please fix this.

[6] I don't understand the methods (described on page 10 of the document for review, which contains lots of introductory pages before the manuscript actually starts) associated with Box D. Since the nodes in the SOM map contain drugs, how are we able to label some of those nodes based on the mutations of tumor cell lines? I _think_ the point is that the "codebook" for each node consists of an average (over the compounds assigned to the node) expression level for each cell line, which is followed by a t-test comparing the expression between cell lines with and without an abnormality. This step seems to be followed by a permutation test to compute empirical p-values. But this means that they have performed more than 4.5 million statistical tests, without making any kind of adjustment for multiple testing. Why should I believe that a cutoff of "p < 0.05" produces anything but random noise in this context? In fact, he may have found *fewer* significant nodes than expected by chance (when you consider that you are performing 368 tests, each with a 5% chance of being picked randomly, per node).

[7] Why was the association between clusters and defective genes performed at the level of 1232 SOM nodes (clades) and not at the level of 28 meta-clades or 7 meta-clade groupings?

[8] The caption for Figure 1 needs to explain/define the numerous abbreviations and acronyms employed. One also suspects that the duplication of "T1" in Box B should actually read "T1, T2".

[9] Figure 2 should include a colorbar to make it easier for readers to interpret the SOM plot. It would help of the author prevented the plot title from overlapping the list of compounds. It would also help to label the subpanels (with something like A, B, C) and to make it clearer why there are two histograms and how they differ. This is especially true since the figure caption does not discuss the histograms. Also, both histograms should use "inter-node" and not the spurious plural.

[10] None of the graphical methods in Figure 3 appear to contain a compelling reason to select the numbers of clusters ("meta-clades") proposed by the author. Nor do they justify the collapse of those categories into seven "meta-clade groupings".

[11] Using a gradient-based color scheme to label discrete meta-clades and meta-clade groupings in Figure 4 is potentially highly misleading. For one thing, it is extremely difficult to tell if the color boundaries actually align with valid branches in the accompanying dendrogram, especially for the gray-scale meta-clade groupings. For visualization, the author might want to consider the Polychrome R package, which can produce a wider variety of distinct colors for displaying discrete classes.

[12] More importantly, he might want to consider other clustering methods (K-means? PAM?) in light of the fact that the hierarchically defined meta-clades frequently break up into distinct subregions of the SOM display, indicating an incompatibility of clusters between the two methods. As sort-of noted by the author, this disparity is probably caused by changing distance metrics. Both hierarchical clustering and PAM can use the exact same distance metric that was applied when performing SOM clustering. And it would be important to use the same distance metric when applying methods to estimate the number of clusters. Visualizing the data with alternative methods such as t-SNE or UMAP might also provide additional insight into whether one should believe the clustering results.

[13] it would be useful to see a version of the original SOM plot (from Figure 2) with the meta-clade boundaries (from Figure 4) superimposed. After all, the original plot indicates the relative distance between neighboring SOM nodes, so it would provide greater support for the clustering if the boundaries tended to run through the yellow areas where neighbors aren't particularly close.

[14] From the figure and its caption, it is not clear what we are supposed to learn from Figure 5. My immediate reaction is that ABL1 is not particularly associated with GI50 values in this SOM node. What I get from this plot is that perhaps one should be using a Wilcoxon rank sum test instead of a t-test to associate defective genes with SOM nodes. I have a similar reaction to Figure 6.

[15] Figure 7 appears to be somewhat misleading by over-representing the amount of data associated with the CellMiner MOA annotations. While the SOM clusters were created from almost 47000 compounds, the MOA data is only available for 104 compounds, which is only about 0.22% of the total compounds considered. Each of those 104 compounds is then apparently used up to eight times. Is there any statistical significance to any of these "interpretive" assignments?

[16] Frankly, I am unclear on exactly how many times Fisher's Exact test was used to associate MOA's with defective genes. Among other things, I have completely lost track of whether these counts are based on SOM nodes, meta-clades, or meta-clade groups. And, once again, there is no sign that any corrections to p-values have been made to account for multiple testing.

[17] Beginning on page 17 of the document for reviewers, the author drifts from presenting results to discussing their interpretation. It would be easier for the reader if these parts of the manuscript remained more clearly separated.

[18] It would be nice if the font sizes in the numerous tables were large enough to be readable.

"This project was funded in whole or in part with Federal funds from the National Cancer Institute, National Institutes of Health, under Contracts No. HHSN261200800001E and HHSN261201700007I."

Reviewers' comments:

Reviewer's Responses to Questions

**Comments to the Author**

1. Is the manuscript technically sound, and do the data support the conclusions?

Reviewer #1: Partly

Reviewer #2: Yes

2. Has the statistical analysis been performed appropriately and rigorously? 

Reviewer #1: No

Reviewer #2: I Don't Know

3. Have the authors made all data underlying the findings in their manuscript fully available?

Reviewer #1: No

Reviewer #2: No

4. Is the manuscript presented in an intelligible fashion and written in standard English?

Reviewer #1: Yes

Reviewer #2: Yes

5. Review Comments to the Author

Reviewer #1: See attached file.

This is filler because the poorly designed input page at Editorial Manager (which is run by people who also don't apparently know how to use the HTTP notion of domains or realms to distinguish logins for different journals) insists on at least 200 character even if you upload a separate file with the full review. We'll see if anyone realizes they can ignore this part.

Reviewer #2: In the manuscript, the author presents work to integrate chemosensitivity of cancer cell lines with their genomic defects and drug mechanisms of FDA-approved agents. The work is an important and useful study, particularly as medical oncology moves towards a targeted, individualized approach to treat each individual’s cancer with a personalized drug combination. Drug re-purposing and drug combinations, along with treatment sequence, all play into this, and the more information we have about how chemosensitivity, genomic alterations and drug action link together, it may be possible to discover how drugs synergize to create a lethal combination to treat many cancers. Knowing the genetic vulnerabilities of an individual’s tumor type is becoming more possible and cost-efficient with the advances in sequencing technologies.

The author extends work by Ikediobi et al. by the inclusion of more screened compounds (~53k vs ~8k) and a larger set of gene mutations (N=365 vs N=24). A novel analysis of self-organizing-maps (SOMs) integrated data from three different sources (NCI60, cBioPortal and CellMiner) and derived links among tumor cell chemosensitivity, genetically defective genes and mechanisms of action (MOA).

While this study is limited to the analysis of data from cancer cell lines, the results are applicable and could be beneficial for pre-clinical studies. This is pointed out in the Discussion as cell lines are not fully representative of human tumors.

Overall the author could improve the structure of the manuscript for readability. For example, the introduction is very brief and ends abruptly with the bullet point list. This should be expanded.

The Methods/Data focuses on describing the boxes in Figure 1. A better approach might be to break this into headings, and then refer to Figure 1, Box A, rather than have Box A as the heading. This is particularly the case when the author goes on to refer to Box B – E. Descriptive headings would be much more useful to the reader.

The figure legends should have titles – what is the result of the figure? Then a description of each panel should follow. In several figures, it would be helpful to increase the size and include panel letters. While many figures have multiple panels, they are referred to as left/right or upper/lower etc. Panel letters would improve clarity. Many figure legends are too brief. Expansion of the figure legends to fully explain the figures would improve readability of the manuscript.

For someone not directly involved in the field, it would be helpful to define terms, i.e. DTP, NSC, etc. This is also true for gene/proteins, like KRAS, etc., and cancer types (TNBC).

There is a misuse of verb tense throughout the manuscript. For example, “Prior analyses find…” – this should be “found”. Many parts are written as if the work is proposed, not completed.

Gene names should be italicized (not bold); human proteins in all capitals, non-italicized. This is standard convention and would improve the clarity when moving between defective genes and their resultant proteins and protein activities.

In some cases, work of others is referred to without the reference, i.e. in Appendix Figure 2 and Holbeck et al. (in the description for Box C).

Figure 2 – it is very difficult to see the drug names that are in white against the colored SOMDTP projections. The MOAs for the drug lists should be specified on the figure. This is listed in the text, but better labelling of the figure would improve clarity.

Figure 3 – the x-axis on each plot is very difficult to see. How is the inflection point of gap statistic and within sum of squares method calculated? Was this a calculation or interpreted from the curves? With this and other figures, more details and primary data used to do these analyses should be included.

Pg 13 – surely there is a reference for nonlinear dimension reduction that is not Wikipedia?

Pg 14/Figure 5 – it would be helpful to have an indication/description of what the cell lines are. This is done for Figure 6; however, it should read, “Here there are 12 tumor cell lines…” not tumor cells.

The clinical significance of the results of the study could be expanded. What will be the next steps using the results obtained?

For Appendix Figures 3 and 4 – is the result of the defective PI3KR1 and IGFR1 surprising?

The headings for Table II could be improved to describe what will follow. More detail could be added to improve readability.

Figures 9 – 14 – each right panel displays partial SOMDTP meta-clades, which are shown fully in Figure 4. It would be helpful to refer back to Figure 4 or superimpose the full SOMDTP for context.

The author refers to ‘defective’ genes throughout; however, it would be useful to point out that mutations, copy number alterations and/or fusion/splice changes can enable and ‘over-activate’ or enhance gene function. Defective implies lack of function, which is not the case with some of the genes mentioned, particularly in a cancer context. Some specific examples do not fall into this (i.e. p. 27 V600E activating mutation of BRAF); however, the author should clearly delineate the difference between defective/lack-of-function versus altered/gain-of-function.

6. PLOS authors have the option to publish the peer review history of their article (what does this mean?). If published, this will include your full peer review and any attached files.

Reviewer #1: **Yes: **Kevin R Coombes

Reviewer #2: No

---

## [Decision Letter · Decision Letter 1]

17 Mar 2021

Bioinformatic Analysis Linking Genomic Defects to Chemosensitivity and Mechanism of Action

PONE-D-20-35751R1

Dear Dr. Covell,

We’re pleased to inform you that your manuscript has been judged scientifically suitable for publication and will be formally accepted for publication once it meets all outstanding technical requirements.

Kind regards,

Rama Krishna Kancha

Academic Editor

PLOS ONE

Additional Editor Comments (optional):

The reviewers approved the revised version for acceptance following the incorporation of suggested changes and addressing their concerns.

Reviewers' comments:

Reviewer's Responses to Questions

**Comments to the Author**

1. If the authors have adequately addressed your comments raised in a previous round of review and you feel that this manuscript is now acceptable for publication, you may indicate that here to bypass the “Comments to the Author” section, enter your conflict of interest statement in the “Confidential to Editor” section, and submit your "Accept" recommendation.

Reviewer #1: All comments have been addressed

Reviewer #2: All comments have been addressed

2. Is the manuscript technically sound, and do the data support the conclusions?

Reviewer #1: Yes

Reviewer #2: Yes

3. Has the statistical analysis been performed appropriately and rigorously? 

Reviewer #1: Yes

Reviewer #2: I Don't Know

4. Have the authors made all data underlying the findings in their manuscript fully available?

Reviewer #1: Yes

Reviewer #2: Yes

5. Is the manuscript presented in an intelligible fashion and written in standard English?

Reviewer #1: Yes

Reviewer #2: Yes

6. Review Comments to the Author

Reviewer #1: (No Response)

Reviewer #2: As the specific analyses used are outside of my area of expertise, I cannot comment on the statistical methods and models employed. This is my shortcoming and not the author's. Importantly, I am very satisfied with the thoughtful and robust response to all of the reviewers' comments. The manuscript has been completely overhauled, and I think the clarity has vastly improved. The topic is of interest to me to inform pre-clinical studies, and I now believe this comes through more clearly with the presentation of the data, improvement in the description of methods used, better organisation of the results and a more thorough discussion. In my opinion, the author has done very interesting work that will be of benefit to pre-clinical experimental design.

7. PLOS authors have the option to publish the peer review history of their article (what does this mean?). If published, this will include your full peer review and any attached files.

Reviewer #1: **Yes: **Kevin R. Coombes

Reviewer #2: No

---

## [Editor Report · Acceptance letter]

22 Mar 2021

PONE-D-20-35751R1 

Bioinformatic Analysis Linking Genomic Defects to Chemosensitivity and Mechanism of Action 

Dear Dr. Covell:

I'm pleased to inform you that your manuscript has been deemed suitable for publication in PLOS ONE. Congratulations! Your manuscript is now with our production department. 

Kind regards, 

on behalf of

Dr. Rama Krishna Kancha 

Academic Editor

PLOS ONE